# Transforming Growth Factor Beta 2 (TGFB2) mRNA Levels, in Conjunction with Interferon-Gamma Receptor Activation of Interferon Regulatory Factor 5 (IRF5) and Expression of CD276/B7-H3, Are Therapeutically Targetable Negative Prognostic Markers in Low-Grade Gliomas

**DOI:** 10.3390/cancers16061202

**Published:** 2024-03-19

**Authors:** Vuong Trieu, Anthony E. Maida, Sanjive Qazi

**Affiliations:** Oncotelic Therapeutics, 29397 Agoura Road, Suite 107, Agoura Hills, CA 91301, USA; vtrieu@oncotelic.com (V.T.); anthony.maida@sapubio.com (A.E.M.)

**Keywords:** biomarker, brain neoplasms, gliomas, prognosis, transforming growth factors, interferon receptors, tumor microenvironment, immunotherapy, methylation, isocitrate dehydrogenase wild-type

## Abstract

**Simple Summary:**

Gliomas develop in the central nervous system from the glial cells that support it. Low-grade gliomas (LGGs) are characterized by a reduced number of immune cells infiltrating the tumor, requiring immune-boosting therapies to release the immunosuppressive environment of the tumor. We conducted a bioinformatics-led study to evaluate the prognostic impact of cell surface receptor markers for tumor-associated macrophages (TAMs) that produce transforming growth factor beta (TGFB) ligands and interferon-gamma receptor activation to maintain the tumor microenvironment in an immunosuppressed state. Detailed investigation of messenger RNA levels and their impact on OS in LGG patients demonstrated that low mRNA levels of a specific TGFB ligand, TGFB2, and the receptor along with the signaling molecules from interferon-gamma receptor activation (IFNGR2, IRF1, IRF5, and STAT1) result in improved survival of LGG patients. LGG patients expressing high levels of TGFB2 and IFNGR2 are over-represented in IDH wild-type tumor samples, suggesting that TGFB2 and IFNGR2 mRNA can be therapeutically targeted in these high-risk patients. Furthermore, demethylation of *TFGB2*, *IFNGR2*, *IRF1*, *IRF5*, *STAT1*, and *CD276* genes also results in worse survival outcomes, pointing to a molecular mechanism that increases the negative prognostic effects of increased mRNA levels of these target genes. We observed that significant upregulation of mRNA expression levels for three TAM markers, MSR1/CD204, CD86, and CD68, suggested that TGFB2 is pivotal in establishing a pro-tumorigenic TME in gliomas mediated through TAMs. Interestingly, a tumor-associated antigen CD276/B7-H3 mRNA expression increased in tumor tissue, giving further insights into the roles of macrophages and tumor cells in the immunosuppressive TME. Therefore, to improve OS in LGG patients, combination therapies must target TGFB2 and IFN-γ activation (via IRF5 inhibition) or immune therapies targeted against CD276/B7-H3.

**Abstract:**

LGG tumors are characterized by a low infiltration of immune cells, requiring therapeutic interventions to boost the immune response. We conducted a study analyzing mRNA expression datasets from the UCSC Xena web platform. To screen for upregulated genes, we sought to compare normal brain tissue with LGG tumor samples. We also used cBioportal to determine the relationship between mRNA expression levels of 513 LGG patients and their overall survival (OS) outcomes. Three tumor-associated macrophage (TAM) markers, MSR1/CD204, CD86, and CD68, exhibited a 6-fold (*p* < 0.0001), 8.9-fold (*p* < 0.0001), and 15.6-fold increase in mRNA expression levels, respectively, in LGG tumors. In addition, both TGFB1 (4.1-fold increase, *p* < 0.0001) and TGFB2 (2.2-fold increase, *p* < 0.0001) ligands were also upregulated in these tumors compared to normal brain tissue, suggesting that TGFB ligands are pivotal in establishing an immunosuppressive, angiogenic, and pro-tumorigenic TME in gliomas mediated through TAMs. In addition, mRNA upregulation of interferon-gamma receptors, IFNGR1 and IFNGR2, and the downstream signaling molecules STAT1, IRF1, and IRF5, pointed to an essential role for IFN-γ mediated remodeling of the TME. Interestingly, the mRNA expression of a tumor-associated antigen, CD276/B7-H3, showed a significant (*p* < 0.0001) 4.03-fold increase in tumor tissue, giving further insights into the roles of macrophages and tumor cells in supporting the immunosuppressive TME. Multivariate Cox proportional hazards models investigating the interaction of TGFB2 and activation of IFNGR2, STAT1, IRF1, or IRF5 showed that the prognostic impact of high mRNA levels (25th percentile cut-off) of TGFB2 was independent of IFNGR2, STAT1, IRF1, or IRF5 mRNA levels (TGFB2^high^ HR (95% CI) = 4.07 (2.35–7.06), 6 (3.62–10.11), 4.38 (2.67–7.17), and 4.48 (2.82–7.12) for models with IFNGR2, STAT1, IRF1, or IRF5, respectively) and age at diagnosis. Patients with high levels of TGFB2 and IFNGR2 were over-represented by LGG patients with isocitrate dehydrogenase wild-type (IDHwt) mutation status. The prognostic impact of high levels of TGFB2 and IDH wild-type observed by the increases in hazard ratios for TGFB2 (HR (95% CI range) = 2.02 (1.05–3.89)) and IDH wild-type (HR (95% CI range) = 4.44 (1.9–10.4)) were independent predictors of survival, suggesting that risk stratification of patients identifies LGG patients with IDH wild-type and high levels of TGFB2 in the design of clinical trials. Furthermore, we have additional IRF5 and CD276/B7-H3 as prognostic markers that can also be targeted for combination therapies with TGFB2 inhibitors. In support of these findings, we demonstrated that low levels of gene methylation in *TGFB2*, *IFNGR2*, *IRF1*, *IRF5*, *STAT1*, and *CD276* were associated with significantly worse overall survival (OS) outcomes. This suggests that potential mechanisms to increase the expression of these prognostic markers occur via the action of demethylation enzymes.

## 1. Introduction

Being the most prevalent type of tumors in the central nervous system (CNS), gliomas originate from supporting glial cells within the CNS. They are classified based on involved cell types, including ependymomas, pilocytic astrocytomas, astrocytomas, oligodendrogliomas, and mixed oligoastrocytomas [1,2]. The survival rate for low-grade gliomas (LGGs) depends on the tumor subtype, extent of resection, and patient age. Studies have shown that patients who undergo a more than 90% extent of resection have a 5-year overall survival rate of 97%, whereas those who undergo a less than 90% extent of resection have a reduced 5-year overall survival rate of 76% [3,4]. With regard to disease progression, up to 56% of patients (141 cases) experienced progression in a 5-year follow-up study [5]. Isocitrate dehydrogenase (IDH) mutation status significantly impacts disease progression, as demonstrated in studies that showed IDH-mutant (IDHmut) tumors having significantly longer progression-free survival (PFS) and overall survival (OS) compared to their IDH wild-type (IDHwt) counterparts [6,7,8,9]. According to recent clinical trial results (NCT04164901), vorasidenib’s ability to penetrate the brain and target mutations in IDH1 and IDH2 enzymes showed significantly improved PFS in patients with grade 2 glioma. The treated group experienced a median PFS of 27.7 months, compared to just 11.1 months for the placebo group [10]. Molecular profiling tools have advanced subtyping of gliomas based on genetic alterations observed in IDHwt LGG tumors regarding the reclassification of gliomas with MRI and histological findings consistent with diffuse astrocytomas grade II and III, whereby these tumors are now categorized as grade 4 tumors with molecular features of glioblastoma based on the presence of TERT promoter mutation, +7/−10, or EGFR amplification, according to the cIMPACT-NOW Update 3 [11,12,13,14]. A group of 157 patients, classified as WHO grade 3 anaplastic astrocytoma (78.3%) and WHO grade 2 diffuse astrocytoma (21.7%), participated in a multicenter study to explore the impact of EGFR amplification and TERT-promotor mutation on overall survival (OS). The patients harbored IDHwt genotype, and the clinical outcomes indicated that both EGFR amplification (*p* = 0.014) and TERT-promotor mutation (*p* = 0.042) were significantly associated with reduced OS [15]. Additionally, next-generation sequencing was conducted on 211 adults with IDHwt gliomas. The analysis identified actionable molecular alterations in 13 of the 211 patients (6.2%), such as BRAF V600E mutation, FGFR3-TACC3 fusion, PIK3CA mutation, and NTRK1 fusion [16]. These observations highlight the importance of using molecular profiling tools for the diagnosis and treatment of IDHwt LGG tumors [17].

Immunotherapies have emerged as a treatment modality for LGG tumors due to their low infiltration of immune cells. Hence, the growth of these tumors is not effectively inhibited by the immune system, necessitating therapeutic interventions to enhance the immune response. Notably, IDHmut and IDHwt gliomas exhibit differences in immune cell infiltration, characterized by the presence of different types of TAMs. In IDHwt gliomas, the tumor microenvironment (TME) creates an immunosuppressive environment that promotes tumor growth and progression [18,19]. IDH mutations exhibit low expression of T-cell markers, enriching CD4+ naive T cells and reducing memory T cells. Comparing IDHmut with IDHwt tumors, the TME upregulates ligands that activate NK receptor recognition, thus promoting NK cell-mediated lysis [20,21,22].

Our analysis employed the use of RNA-seq expression and methylation data to understand the role of tumor-associated macrophages (TAMs) in relation to OS outcomes in LGG patients. TAMs are generally categorized into M1 and M2 functional classes, and M2-like macrophages comprise a significant proportion of immune cells in gliomas that are associated with an immune-suppressive tumor microenvironment (TME). Sub-classifications of M2-like macrophages are based on their roles in diverse responses of the immune system: M2a macrophages contribute to tissue repair and regeneration, M2b macrophages are involved in wound healing and angiogenesis, M2c macrophages regulate anti-inflammatory response, and M2d macrophages contribute to tissue remodeling and angiogenesis. Two subtypes of M2-like macrophages, M2c and M2d, are responsible for producing TGFB ligands that maintain the TME in an immunosuppressive state [23,24]. These subtypes of M2 macrophages also regulate the expression of extracellular matrix proteins, which augment tumor migration and invasion [25,26,27]. One study found that TGFB2, through MMP-2 expression, supported tumor cell invasion by breaking down the basal membrane [28]. Furthermore, TGF-β signaling can regulate glioma invasion through the TGF-β1 signaling pathway, with glioma stem-like cells (GSLCs) exhibiting upregulation of TGF-β receptor 2 for both mRNA and protein [29]. In addition to the production of TGFB ligands to influence the immune-suppressive TME, interferon-gamma (IFN-γ) production modulates the remodeling of the TME. IFN-γ can affect the expression of programmed death-ligand 1 (PD-L1) in EBV-positive gastric carcinoma by activating the JAK2/STAT1/IRF-1 signaling pathway [30]. IRF-1 was found to be a predictive prognostic marker for anti-PD-1 therapy in metastatic melanoma [31]. The expression levels of RUNX1 and IFNGR2 were significantly reduced, and their correlation was enhanced in the IDHmut subtype of LGG. Patients with high RUNX1 and/or IFNGR2 expression in the IDHmut subtype showed poorer prognosis and significantly different immune infiltration patterns [32].

Given the important roles of TGFB ligands and IFN-γ released by immune cells in the TME as related to IDH mutational status, we tested the clinical prognostic significance of high tumor TGFB2 and IFNGR2 mRNA levels measured in low-grade gliomas. In addition, we also determined that known downstream signaling molecules activated by interferon-gamma receptors associated with macrophages, namely STAT1, IRF1, and IRF5, were also correlated with poor OS outcomes in tumor samples. Tumors with high levels of expression of TGFB2 and in combination with high levels of IFNGR2, IRF1, IRF5, CD86, CD68, MSR1/CD204, and CD276 exhibited negative prognosis for OS outcomes, providing for new modes of therapy for low-grade gliomas that are capable of knocking down TGFB2 expression in combination with either IRF5 blockade or targeting CD276/B7-H3. Multivariate Cox regression models indicated that high levels of TGFB2 mRNA expression could serve as a potential biomarker for LGG patients independent of age, expression of IFNGR2/STAT1/IRF1/IRF5, and the interaction effect of TGFB2 with these markers. We observed that the subset of patients with high levels of TGFB2 and IFNGR2 was also over-represented by tumors with IDHwt genotypes. Even in this context, high TGFB2 mRNA levels were an independent predictor for worse OS in a multiple regression Cox model that included age, IDH mutational status, and an interaction term for TGFB2 × IDH mutational status. These observations were further supported by examining the impact on OS of *TGFB2*, *IFNGR2*, *IRF1*, *IRF5*, *STAT1,* and *CD276* gene methylation levels, whereby genes with low levels of methylation exhibited significantly worse OS outcomes, suggesting that one mechanism by which expression of these negative prognostic markers could be increased is through the action of demethylation enzymes.

## 2. Materials and Methods

### 2.1. Comparing mRNA Expression in LGG Tumors and Normal Brain Samples

Expression metric summarized as log2 transformed transcripts per million (TPM) downloaded as RNAseq data files (https://toil-xena-hub.s3.us-east-1.amazonaws.com/download/TcgaTargetGtex_rsem_gene_tpm.gz; full metadata, accessed on 25 July 2023) from the UCSC Xena web platform (https://xenabrowser.net/datapages/ accessed on 25 July 2023) [33] were used to compare mRNA levels for 509 evaluable LGG patients versus 1141 brain samples from all regions of the brain enabled by the standardized realigned and recalculated gene and transcript expression data set for all TCGA, TARGET, and GTEx samples (UCSC Toil RNAseq recompute compendium) [34]. We contrasted mRNA expression between TCGA “tumor” samples and corresponding GTEx “normal” samples.

### 2.2. Kaplan–Meier OS Analysis Comparing Subsets of LGG Patients Stratified According to mRNA Expression Levels of TGFB2 and Target Genes in the TME

Clinical metadata and mRNA expression data obtained from 513 LGG patients were utilized to compare OS outcomes for mRNA level stratification of gene tumors (https://www.cbioportal.org/study/summary?id=lgg_tcga_pan_can_atlas_2018; accessed 20 April 2023).

Demographics of these 513 LGG patients revealed that 285 were male and 228 were female. The mean (±SEM) and median (range) of the age distribution were 42.9 ± 0.6 and 41 (14–87) years, respectively. These LGG patients were treated with radiation (N = 299), temozolomide (N = 259; alkylates/methylates DNA), bevacizumab (N = 48; humanized monoclonal antibody that blocks angiogenesis by inhibiting vascular endothelial growth factor A (VEGF-A)), and lomustine (N = 39; a cell cycle non-specific, highly lipophilic alkylating agent). According to the mutational status of the LGG patients, 167 were identified as IDH mutants with chromosome 1p and 19q co-deletion (LGG_IDHmut-codel), while 247 were reported as IDH mutant, 92 were recorded as IDHwt, and 7 were marked as NA.

Data files harboring TPM expression values were downloaded from the cBioportal as a .tsv file and were processed to calculate the ranked quartiles of gene expression from 513 LGG patients. Patient groupings were sub-divided into 4 groups based on their expression levels of TGFB2 and Gene2 (genes coding for mRNA expressed in the TME included TGFB1, TGFB2, TGFB3, IFNGR1, IFNGR2, STAT1, IRF1, IRF5, CD68, CD86, CD276, and MSR1/CD204): TGFB2^high^/Gene2^high^ (higher than or equal to the 25th percentile of both TGFB2 and Gene2), TGFB2^low^/Gene2^low^ (lower than the 75th percentile of both TGFB2 and Gene2), and combinations of high and low expression levels for both genes under investigation (TGFB2^high^/Gene2^low^, and TGFB2^low^/Gene2^high^). After censoring the OS curves at 120 months, we compared the survival impact of the combinations of TGFB2 and Gene2 levels on OS. We also evaluated the impact of either TGFB2^high^ or Gene2^high^ versus the remaining patients on OS to test the effect of the subset of patients with high gene expression levels. OS outcomes were determined using the Kaplan–Meier (KM) method tested for statistical significance (log-rank Chi-square test, implemented utilizing R-based software packages (version 4.1.2), including survival_3.2-13, survminer_0.4.9, and survMisc_0.5.5. To present the treatment outcomes in a graphical format, we plotted graphs using dplyr_1.0.7, ggplot2_3.3.5, and ggthemes_4.2.4 implemented in R). We considered *p*-values less than 0.05 significant after adjusting for multiple comparisons across four groups (6 comparisons) using the Benjamini and Hochberg method.

### 2.3. Comparing the Independent Effects on Hazard Ratios (HR) of High Levels of TGFB2 mRNA Levels Controlling for Target Gene Expression Levels, Age, and Interaction between TGFB2 and the Target Gene

We assessed the statistically independent effects of TGFB2 and Gene2 levels on OS by performing multivariate analyses using the multivariate Cox proportional hazards model. We controlled for age at diagnosis and the interaction between TGFB2 and Gene2. Briefly, the model included (i) the mRNA expression level for TGFB2 as a categorical variable comparing high versus low TGFB2 mRNA expression levels at 25% cut-off for expression values, (ii) Gene2 mRNA expression as a categorical variable comparing high versus low Gene2 mRNA expression levels (25% cut-off), and (iii) age at diagnosis implemented in R (survival_3.2-13 ran in R version 4.1.2.). Forest plots were utilized to visualize the effect sizes of the hazard ratios for Cox proportional hazards models (survminer_0.4.9 ran in R version 4.1.2 (1 November 2021). The life table hazard ratios (HRs) were estimated using the exponentiated regression coefficients implemented in R (survival_3.2-13 ran in R version 4.1.2). We conducted an analysis that investigated the effect of adding an interaction term (TGFB2 × Gene2) to determine gene expression dependencies in the model.

### 2.4. Correlation of TGFB2 and Immune Cell Infiltration

Estimations of the correlation between TGFB2 and macrophage immune-cell infiltration in LGG tumors were calculated using the algorithms provided in the TIMER2.0 (http://timer.cistrome.org/ accessed on 28 July 2023) web tool compiled for the Cancer Genome Atlas (TCGA). The web portal calculates and provides bivariate plots for the purity-adjusted Spearman’s rho correlations utilizing CIBERSORT-ABS, QUANTISEQ, and XCELL immune deconvolution methods to estimate infiltration of M2 and M1 macrophages in LGG tumors [35,36].

### 2.5. OS Impact in LGG Patients upon Demethylation of the Target Genes

We analyzed clinical metadata and methylation beta values obtained from HumanMethylation450 (HM450) arrays for 513 patients who were diagnosed with LGG (https://www.cbioportal.org/study/summary?id=lgg_tcga data file: “data_methylation_hm450.txt” accessed on 13 January 2024). Compared to the remaining patients, we investigated the OS impact of reduced methylation of the target genes *TGFB2, IFNGR2, IRF5, IRF1, STAT1,* and *CD276* (25th percentile cut-off for each gene). We tested whether demethylation of the target genes was negatively prognostic at high levels of mRNA expression, resulting in worse OS outcomes.

## 3. Results

### 3.1. LGG Tumors Exhibit Augmented mRNA Expression Levels for TGFB Ligands, Interferon-Gamma Receptors, Downstream Signaling Molecules, and Macrophage Markers Compared to Normal Brain Samples

Comparison of mRNA expression levels in LGG tumor (509 patients evaluated in the Xena portal) versus normal tissues showed significant upregulation (*p* < 0.0001) in tumors for ligands TGFB1/2, interferon-gamma activated receptors, and the corresponding downstream signaling molecules (IFNGR1, IFNGR2, STAT1, IRF1, and IRF5), and markers for tumor-associated markers (TAMs) MSR1 and CD86 exhibited very low expression levels in normal tissue (<0 log2 TPM equivalent to TPM value of 1) but experienced a significant increase in expression levels in tumor tissue. For patients with LGG, the expression of MSR1 mRNA was 2.3 ± 0.07, whereas in normal tissue, it was −0.27 ± 0.06, a 6-fold increase (*p* < 0.0001). In normal tissue, CD86 mRNA expression was −0.74 ± 0.06, whereas in LGG patients, it was 2.41 ± 0.06, an 8.9-fold increase (*p* < 0.0001). Investigation of the expression of TGFB isoforms showed overexpression of TGFB1 and TGFB2 in LGG tumors but not for TGFB3 (1.1-fold increase in LGG tumors compared to normal tissue, *p* = 0.1). The TGFB1 mRNA expression in normal tissue had a mean of 2.45 ± 0.04, whereas it was 4.5 ± 0.04 in LGG patients, showing a significant (*p* < 0.0001) 4.1-fold increase. The expression of TGFB2 mRNA showed an average (±SEM) of 1.48 ± 0.04 for healthy tissue and 2.64 ± 0.08 for patients with LGG, representing a significant (*p* < 0.0001) 2.2-fold increase. The expression of IFNGR1 mRNA was found to be 4.33 ± 0.04 in normal tissue and 5.98 ± 0.03 in LGG patients, representing a significant (*p* < 0.0001) 3.1-fold increase. Similarly, the mRNA expression of IFNGR2 was measured in normal tissue and LGG patients. The mean expression in normal tissue was 4.21 ± 0.04, whereas in LGG patients, it was 5.17 ± 0.03, showing a significant (*p* < 0.0001) 1.9-fold increase in mRNA expression (Figure 1, Appendix A).

### 3.2. Expression Levels of TGFB2, IFNGR2, CD276, and MSR1/CD204 mRNA Were Treatment-Independent Negative Prognostic Markers in LGG Tumors

We screened tumor and immune cell mRNA expression of cell surface receptors and signaling molecules comparing high expression of these molecules (upper 25% percentile) versus remaining LGG patients for all the patients in the TCGA dataset (N = 513) using a univariate Cox proportional hazards model for patients treated with temozolomide (N = 259), radiation (N = 299), bevacizumab (N = 48), or lomustine (N = 39). Expression of TGFB2, IFNGR2, CD276, and MSR1/CD204 exhibited significant increases in hazard ratios in all treatment protocols (Appendix A). Patients with the highest levels of TGFB2 mRNA expression showed the most significant increases in hazard ratios for all patients (HR (95% CI) = 3.92 (2.74–5.61)) and patients treated with temozolomide (4.01 (2.61–6.17)), radiation (3.56 (2.38–5.34)), and bevacizumab (4.42 (2.1–9.29)).

The differences between IDHwt and mutant were also explored, with the following findings.

### 3.3. TGFB2 Is a Significant Negative Prognostic Indicator for OS in LGG Patients Independent of IDH Mutational Status

Upon analyzing the correlation between IDH mutational status and overall survival time, comparing 92 IDHwt LGG patients with the remaining 421 patients showed that the median OS time for the IDHwt group was 21 months (95% CI: 17.7–25.5 months; N = 92; 50 events), which was significantly lower than the 421 remaining patients (OS = 98 months (95% CI: 80–134 months; 75 events; log-rank Chi-square = 63.3, *p* < 0.0001). There was a significant over-representation of patients with high TGFB2 in the IDHwt group of patients compared to the remaining patients, whereby 51 out of 421 remaining patients (12%) had high TGFB2 expression, whereas 78 out of 92 IDHwt patients (85%) had high TGFB2 expression (odds ratio = 40, Fisher’s exact test, *p* < 0.0001)). We also observed a significant over-representation of patients expressing high levels of IFNGR2 in IDHwt TGG patients, whereby 65 out of 92 IDHwt patients (71%) exhibited high IFNGR2 expression compared to 64 out of 421 remaining patients (15%) (odds ratio = 13.3, Fisher’s exact test, *p* < 0.0001)) (Appendix A).

Using the multivariate Cox proportional hazards models, we evaluated the independent effects of TGFB2 or IFNGR2 and IDHwt mutational status on OS (Appendix A). The hazard ratio (HR) for patients in the TGFB2^high^ group showed a significant increase, with an HR (95% CI range) of 2.02 (1.05–3.89) (*p* = 0.036) referenced to the remaining group of patients. Furthermore, the IDHwt group of patients also exhibited a significant increase in HR, with an HR (95% CI range) of 4.44 (1.9–10.4); *p* = 0.001) controlling for age at diagnosis with an HR (95% CI range) of 1.05 (1.03–1.06) (*p* < 0.001). Similarly, we observed that patients in the IFNGR2^high^ group exhibited a significant increase in HR (HR (95% CI range) = 2.04 (1.16–3.57); *p* = 0.013), and patients in the IDHwt group also showed a significant increase in HR (HR (95% CI range) = 3.52 (1.81–6.84); *p* < 0.001), controlling for the age at diagnosis (HR (95% CI range) = 1.05 (1.03–1.07); *p* < 0.001) (Appendix A). These results suggest an increase in the HR observed for TGFB2^high^ and IFBGR2^high^ patients, independent of age and IDHwt mutational status in LGG patients (Appendix A).

### 3.4. TGFB2 and IFNGR2 Are Significant Negative Prognostic Indicators for OS in LGG Patients

Our objective was to assess the prognostic significance of gene expression levels and OS in LGG patients for TGFB2 (Figure 2A), IFNGR2 (Figure 2B), and IFNGR1 (Figure 2C) by comparing high- and low-expressing subsets of 513 LGG patients (25th percentile cut-off). A group of patients with high TGFB2 levels (TGFB2^high^; N = 129, 55 events) exhibited significantly shorter overall survival times (median = 27 months, 95% CI = 23.7–54.8 months) compared to those with low levels (TGFB2^low^; N = 384, 70 events, median = 98 months, 95% CI = 80–139 months; log-rank Chi-square of 63.31, *p* < 0.0001) (Figure 2A). Another subset of patients with high IFNGR2 levels (IFNGR2^high^; N = 129, 55 events) also had significantly shorter overall survival times (median = 34 months, 95% CI = 25.5–63.5 months) compared to those with low mRNA levels of IFNGR2 (IFNGR2^low^; N = 384, 70 events, median = 98 months, 95% CI = 87.5–139 months; log-rank Chi-square of 52.06, *p* < 0.0001) (Figure 2B). The OS outcome for the IFNGR1^high^ subset of patients (median = 75, 95% CI = 63.5–130.8 months, N = 129, 35 events) was not statistically different from the OS outcome in the IFNGR1^low^ subset of patients (median = 95; 95% CI = 63–145 months, N = 384, 90 events, log-rank Chi-square = 0.03, *p* = 0.867) (Figure 2C).

### 3.5. Interferon-Gamma Receptor-Activated Downstream Signaling Molecules STAT1, IRF1, IRF5 Displayed Significant Negative Prognosis for OS in LGG Patients

Interferon-gamma receptor-activated downstream signaling molecules also exhibited negative prognostic outcomes at increased levels of mRNA expression. Patients with high levels of IRF5 (IRF5^high^; median = 64 months; 95% CI = 40.8–NA months, N = 129, 42 events) had significantly shorter overall survival (OS) times compared to those with low levels of IRF5 (IRF5^low^; median = 95 months; 95% CI = 75–134 months; N = 384, 83 events, log-rank Chi-square = 10.55, *p* = 0.001) (Figure 3A). Similarly, patients with high levels of IRF1 (IRF1^high^; median = 49 months; 95% CI = 30.2–NA months; N = 129, 46 events) exhibited significantly shorter OS times compared to those with low levels of IRF1 (IRF1^low^; median = 98 months; 95% CI = 78.2–134 months, N = 384, 79 events, log-rank Chi-square = 30.11, *p* < 0.0001) (Figure 3B). Additionally, patients with high levels of STAT1 (STAT1^high^; median = 51 months, 95% CI = 34–75 months, N = 129, 51 events) exhibited significantly worse OS times compared to those with low levels of STAT1 (STAT1^low^; median = 98 months, 95% CI = 87.5–145.1 months, N = 384, 74 events, log-rank Chi-square = 27.62, *p* < 0.0001) (Figure 3C).

### 3.6. The Prognostic Impact of TGFB2 mRNA Levels in Tumors of LGG Patients Is Dependent on IFNGR2 mRNA Expression

We determined the prognostic influence of TGFB2 mRNA levels in the context of high and low levels of IFNGR2 mRNA levels and the corresponding mRNA coding for signaling proteins IRF1/5 and STAT1 by stratifying patients into four groups with combinations of high and low expression of both genes. At low levels of IFNGR2, TGFB2^high^ patients exhibited worse OS compared to TGFB2^low^ patients (median = 26 and 98 months, respectively; *p* < 0.0001). At high levels of IFNGR2, the TGFB2^high^ group of patients exhibited shorter survival times, but this reduction was not statistically significant (median = 27 versus 64 months; *p* = 0.15). At low levels of TGFB2 mRNA levels, IFNGR2^high^ group of patients exhibited worse survival outcomes compared to IFNGR2^low^ patients (median = 64 vs. 98 months respectively; *p* < 0.0001)) (Figure 4).

### 3.7. The Prognostic Effect of TGFB2 mRNA Levels in Tumors of LGG Patients Is Independent of the Levels of IRF5

Examining patients with high and low combinations of TGFB2 and IRF5 expression showed that the median survival time for 316 patients in expression group TGFB2^low^/IRF5^low^ was 105 months, with 53 recorded events and a 95% confidence interval of 80–120 months (Figure 5). For the group of TGFB2^low^/IRF5^high^ patients, the median survival time for 68 patients was 93 months, with 17 recorded events and a 95% confidence interval of 63.5 to NA months. In the group of TGFB2^high^/IRF5^low^ patients, the median survival time for 68 patients was 26 months, with 30 recorded events and a 95% confidence interval of 21 to NA months. Lastly, for the expression group TGFB2^high^/IRF5^high^, the median survival time for 61 patients was 27 months, with 25 recorded events and a 95% confidence interval of 23.9 to NA months. At lower expression levels of IRF5 mRNA expression, there was a significant difference between the TGFB2^low^/IRF5^low^ and TGFB2^high^/IRF5^low^ groups of patients (105 compared to 26 months, respectively; *p* < 0.0001)). At higher levels of IRF5 mRNA expression, there was a significant difference between low and high levels of TGFB2 expression (TGFB2^low^/IRF5^high^ versus TGFB2^high^/IRF5^high^ (93 compared to 27 months, respectively; *p* = 0.0007)) (Figure 5).

### 3.8. The Prognostic Impact of TGFB2 mRNA Levels in Tumors of LGG Patients Is Independent of the Levels of IRF1, While at TGFB2^low^, IRF1 Exhibits Worse OS

We examined the impact on OS for patients expressing high and low combinations of TGFB2 and IRF1. A total of 329 patients in the TGFB2^low^/IRF1^low^ group (Figure 6) exhibited a median survival time of 105 months (95% CI = 93.2–120, 55 events). The TGFB2^low^/IRF1^high^ group was composed of 55 patients, with 15 death events and a median survival time of 64 months (with a 95% CI of 39–NA). The TGFB2^high^/IRF1^low^ group experienced 24 death events among 55 patients, resulting in a median survival time of 26 months (with a 95% CI of 21.3–NA). Lastly, out of 74 patients in the TGFB2^high^/IRF1^high^ group, 31 experienced death events and exhibited a median survival time of 29 months (with a 95% CI of 23.7–NA). We revealed that at IRF1^low^ mRNA levels, TGFB2^low^ patients experienced longer OS than TGFB2^high^ patients (105 versus 26 months, respectively; *p* < 0.0001). Also, comparing IRF1^high^ groups of patients, it was shown that patients with high levels of TGFB2 experienced shorter OS times compared to TGFB2^low^ (29 versus 64 months, respectively; *p* = 0.046) than patients expressing low levels of TGFB2 mRNA. At TGFB2^low^ mRNA levels, IRF1^high^ patients exhibited significantly shorter OS times than IRF1^low^ patients (64 vs. 105 months, respectively; *p* = 0.0011) (Figure 6).

### 3.9. At Low Levels of STAT1, High TGFB2 mRNA Levels Exhibit Worse Survival Outcomes, and at Low Levels of TGFB2, STAT1 Is a Significant Negative Prognostic Indicator for OS in LGG Patients

Patients with low expression of both TGFB2 and STAT1, TGFB2^low^/STAT1^low^, exhibited a median survival time of 114 months, with 49 recorded events and a 95% confidence interval of 94.5–120 months. LGG patients in the TGFB2^low^/STAT1^high^ group had a median survival time of 51 months, 21 events, and a 95% confidence interval of 38.9–NA. For the TGFB2^high^/STAT1^low^ group of LGG patients, the median survival time was 27 months, with 25 recorded events and a 95% confidence interval of 21.3–NA. Lastly, the TGFB2^high^/STAT1^high^ group of patients exhibited a median survival time of 27 months, experiencing 30 events and a 95% confidence interval of 23.7–NA (Figure 7). Investigation of TGFB2 and STAT1 levels showed that at low TGFB2 mRNA levels, patients with expression levels in the upper quartile for STAT1 experienced significantly shorter survival times (114 versus 51 months; *p* < 0.0001). At low STAT1 mRNA levels, patients expressing high levels of TGFB2 experienced worse survival outcomes (114 vs. 27 months; *p* < 0.0001). Patients expressing the lowest levels of TGFB2 and STAT1 showed more favorable OS outcomes than patients expressing the highest levels of both TGFB2 and STAT1 (114 vs. 27 months; *p* < 0.0001) (Figure 7).

### 3.10. Independent Impact on OS of High TGFB2, IFNGR2 mRNA, and Downstream Signaling Molecules IRF5, IRF1, and STAT1 Determined from Multivariate Cox Proportional Hazards Models

Using the multivariate Cox proportional hazards model, we demonstrated that the effect of TGFB2 and IFNGR2 on OS was statistically independent, controlling for the effect of age at diagnosis and the interaction term. The TGFB2^high^ group of patients (HR (HR (95% CI range) = 4.07 (2.35–7.06); *p* < 0.001)) and IFNGR2^high^ patients (HR (95% CI range) = 3.29 (1.91–5.65); *p* < 0.001) exhibited significant increases in HR compared to low TGFB2 or IFNGR2 groups, respectively (Figure 8A). Similarly, significant independent effects of the TGFB2^high^ group of LGG patients who displayed a significant increase in HR (HR (95% CI range) = 4.48 (2.82–7.12); *p* < 0.001)) were independent of the effect observed for the IRF5^high^ group of patients showing an increase in HR (HR (95% CI range) = 1.94 (1.12–3.37); *p* = 0.018). The hazard ratio was more than double that of the independent effect of TGFB2 compared to IRF5, controlling for age at diagnosis and the interaction term (Figure 8B). Both TGFB2 (HR (95% CI range) = 4.38 (2.67–7.17); *p* < 0.001) and IRF1 (HR (95% CI range) = 3.42 (1.9–6.15); *p* < 0.001) demonstrated independent increases in OS (Figure 8C). In the multivariate Cox proportional hazards model that included TGFB2 and STAT1, patients in the TGFB2^high^ expression group exhibited a significant increase in HR (HR (95% CI range) = 6 (3.62–10.11); *p* < 0.001). Similarly, patients in the STAT1^high^ group also displayed a significant increase in HR (HR (95% CI range) = 2.93 (1.75–4.91); *p* < 0.001) (Figure 8D).

### 3.11. TGFB2 mRNA Levels Were Positively Correlated with M1 and M2 Macrophage Infiltration into LGG Tumors

We estimated the correlation of TGFB2 and macrophage immune-cell infiltration in LGG tumors using the algorithms provided in TIMER2.0 (http://timer.cistrome.org/, accessed on 26 August 2023). These algorithms that calculated Spearman’s rho correlations utilizing CIBERSORT-ABS, QUANTISEQ, and XCELL immune deconvolution methods showed that TGFB2 expression values (RSEM estimated TPM values) were significantly positively correlated with both M1 and M2 macrophages using the three deconvolution methods (Appendix A).

Further examination of subsets of patients that expressed high levels of both TGFB2 and IRF5 or IRF1 with respect to molecular subtypes and OS outcomes are depicted in Figure 9. Out of the 61 LGG patients in the TGFB2^high^/IRF5^high^ subset of patients, 36 (59%) were classified as IDHwt genotype, and the survival outcome of these 61 patients was significantly worse than the remaining patients (TGFB2^high^/IRF5^high^ median = 27 months (95% CI = 23.9–NA) vs. the remaining group median OS = 95 months (95% CI = 75–120 months; log-rank Chi-square = 28.97, *p* < 0.001)) (Figure 9A). Out of the 74 patients in the TGFB2^high^/IRF1^high^ subset of patients, 48 (65%) were classified as IDHwt and the OS times exhibited a significant difference compared to the remaining patients (TGFB2^high^/IRF1^high^ median = 29 months (95% CI: 23.7–NA) vs. remaining patients median = 95 months (95% CI: 75–120; log-rank Chi-square = 34.3, *p* < 0.001)) (Figure 9B). Next, we examined the negative prognostic cell markers for TAMs expressed at high levels of TGFB2, and CD68, MSR1, CD276, and CD86 cell markers exhibited worse OS outcomes (Figure 9C–F). The TGFB2^high^/CD68^high^ group of 60 patients exhibited a median OS time of 34 months (95% CI = 23.7–NA months), which was significantly shorter than the OS for the remaining group of 453 patients with a median OS time of 95 months (95% CI = 75–120 months; log-rank Chi-square value = 24, *p* < 0.0001) (Figure 9C). The 76 patients with TGFB2^high^ and MSR1^high^ mRNA expression exhibited a median OS time of 27 months (95% CI = 23.7–63 months), which was significantly shorter than the median OS time for the remaining 437 patients of 96 months (95% CI of 78.2–120 months; log-rank Chi-square = 46.52, *p* < 0.0001) (Figure 9D). There was a significant difference in the survival rates among the groups comparing the TGFB2^high^/CD276^high^ arm of the OS curve (median OS time for 76 patients was 27 months (95% CI = 23.7–63)) with the remaining 437 patients (median OS time of 96 months; 95% CI = 78.2–120) and 89 recorded events; log-rank Chi-square = 38.6, *p* < 0.0001) (Figure 9E). The survival outcomes between TGFB2^high^/CD86^high^ and the remaining patients were found to be statistically significantly different (log-rank Chi-square = 20.2, *p* < 0.0001). Among 60 patients in the TGFB2^high^/CD86^high^ group, the median OS time was 34 months (95% CI = 26–NA, and 23 events). For the remaining group, which consisted of 453 patients, the median OS time was 95 months (95% CI = 73.5–120) (Figure 9F).

### 3.12. Demethylation of Target Genes TGFB2, IFNGR2, IRF5, IRF1, STAT1, and CD276 Exhibited Significantly Shorter OS Times and Overrepresentation in IDH Wildtype for LGG Patients

Figure 10A shows that the median OS time for 129 patients with low levels of methylation for the *TGFB2* gene was 25.5 months (95% CI: 21.3–37.9, events = 61) and significantly shorter than for the 384 remaining patients (median OS = 105.2 months (95% CI: 80–139), events = 64; log-rank Chi-square = 96.5, *p* < 0.001). Patients with low levels of *TGFB2* methylation were significantly over-represented in the patients with IDHwt genotype (92/92 = 100%) compared to 37 *TGFB2*^lowMe^ patients with IDH mutations (37/421 = 8.8%; Fisher’s exact test, 2-tailed, *p* < 0.0001). The median OS time for LGG patients with low levels of methylation for the *IFNGR2* gene was 40.8 months (95% CI: 33.2–75.2, events = 49) and was significantly shorter than the median OS time for the remaining patients (median OS = 98.2 months (95% CI: 80–134.2), events = 76; log-rank Chi-square = 39.2, *p* < 0.001). Patients with low levels of *IFNGR2* methylation were significantly over-represented in the patients with IDHwt genotype (64/92 = 69.6%) compared to 65 *IFNGR2*^lowMe^ patients with IDH mutations (65/421 = 15.4%; *p* < 0.0001) (Figure 10B). The median OS time for 129 patients with low levels of methylation for the *IRF5* gene was 48.7 months (95% CI: 37.9–94.5, events = 48) and was significantly shorter than the median OS time for the remaining patients (median OS = 98.2 months (95% CI: 80–139), events = 77; log-rank Chi-square = 23.03, *p* < 0.001). Patients with low levels of *IRF5* methylation were significantly over-represented in the patients with IDHwt genotype (52/92 = 56.5%) compared to 77 *IRF5*^lowMe^ patients with IDH mutations (77/421 = 18.3%; *p* < 0.0001) (Figure 10C). The median OS time for 129 patients with low levels of methylation for the *IRF1* gene was 34 months (95% CI: 26.8–48.7, events = 55) and was significantly shorter than the median OS time for the remaining patients (median OS = 98.2 months (95% CI: 80–139), events = 70; log-rank Chi-square = 58.5, *p* < 0.001). Patients with low levels of *IRF1* methylation were significantly over-represented in the patients with IDHwt genotype (68/92 = 73.9%) compared to 61 *IRF1*^lowMe^ patients with IDH mutations (61/421 = 14.5%; *p* < 0.0001) (Figure 10D). The median overall survival time for 129 patients with low levels of methylation for the *STAT1* gene was 40.1 months (95% CI: 31.4–75.2), events = 53) and was significantly shorter than the median OS for the remaining patients (median OS = 98.2 months (95% CI: 80–139), events = 72; log-rank Chi-square = 32.8, *p* < 0.001). Patients with low levels of *STAT1* methylation were significantly over-represented in the patients with IDHwt genotype (55/92 = 59.8%) compared to 74 *STAT1*^lowMe^ patients with IDH mutations (74/421 = 17.6%; *p* < 0.0001) (Figure 10E). The median OS time for 129 patients with low levels of methylation for the *CD276* gene was 25.5 months (95% CI: 21.3–37.9, events = 60) and was significantly shorter than the median OS time for the remaining patients (median OS = 95.6 months (95% CI: 80–133.7), events = 65; log-rank Chi-square = 83.7, *p* < 0.001). Patients with low levels of *CD276* methylation were significantly over-represented in the patients with IDH-wildtype genotype (89/92 = 96.7%) compared to 40 *CD276*^lowMe^ patients with IDH mutations (40/421 = 9.5%; *p* < 0.0001) (Figure 10F).

## 4. Discussion

### 4.1. Differential Expression of Macrophage Markers, TGFB Ligands, Interferon-Gamma Receptor, and Signaling Molecules in LGG Tumors Compared to Normal Brain Tissue

The functioning TAMs in the TME are affected by signaling mechanisms present in the TME. The TAMs have a dual role to play in tumor progression, as they can either promote or suppress tumor growth. The current model categorizes TAMs into two phenotypes—the M1 phenotype (anti-tumor) and the M2 phenotype (pro-tumor), based on in-vitro observations of cell function [23,24,37,38]. However, this classification does not accurately reflect the situation in vivo, as individual markers for M1-like and M2-like TAMs show both pro-tumor and anti-tumor effects. Additionally, individual TAMs in human gliomas have been found to co-express both canonical M1 and M2 markers, indicating diverse phenotypes for TAMs in the TME [39,40]. The expression of two TAM markers in our studies revealed that macrophage scavenger receptor 1 (MSR1/CD204) and CD86 were expressed at very low levels in normal tissue (less than 0 log2 TPM; <1 TPM) and exhibited increased levels in tumor tissue. MSR1 demonstrated a 6-fold increase (mRNA expression of 2.3 ± 0.07, *p* < 0.0001) and CD86 showed an 8.9-fold increase in expression (mRNA expression of 2.41 ± 0.06, *p* < 0.0001) in LGG tumors. Expression of the M2-like TAM marker MSR1/CD204 [37] was found to be an independent prognostic factor for LGG patients and may play a vital role in their immunotherapy [41]. CD86 is expressed on M1-like and M2b-like macrophages [23,37] and may contribute to an immunosuppressive TME through activation of CD4+FoxP3+ regulatory T cells (Treg) [42]. Expression of another TAM marker, CD68 [43], exhibited the second highest expression value in the LGG tumors that also exhibited the greatest fold change in expression (mRNA expression in normal tissues = 1.97 ± 0.06, whereas expression in LGG patients was 5.93 ± 0.06, marking a significant (*p* < 0.0001) 15.6-fold increase in mRNA expression in tumors). A previous study investigated the role of CD68 in glioma at the transcriptome level. It demonstrated its correlation with OS in 325 RNA-seq data from the Chinese Glioma Genome Atlas (CGGA) and 697 RNA-seq data from The Cancer Genome Atlas (TCGA) network. The study revealed that CD68 expression was positively correlated with the malignancy grade of glioma. High CD68 expression was associated with shorter OS times and was predominantly expressed in IDHwt and mesenchymal subtype, indicating its close relationship with inflammatory and immune responses. CD68 was also found to be a specific marker for macrophages in the inflammatory response and played a crucial role in suppressing T-cell-mediated immunity [44] (Figure 1, Appendix A). Significant upregulation was observed for the mRNA expression of B7-H3 (CD276), a member of the B7 family that plays an immunoregulatory role in the T-cell response, highlighting this molecule as a novel potential target for cancer immunotherapy [45,46]. Statistical contrast of normal tissue and primary tumor for CD276 mRNA expression in our study showed a significant (*p* < 0.0001) 4.03-fold increase in mRNA expression relative to normal tissue (Figure 1, Appendix A).

TGFB1 and TGFB2 have been identified as the most highly expressed tumor-promoting cytokine in the TME of gliomas [47,48,49,50,51,52], prompting us to characterize the abundance of TGFB1 and TGFB2 mRNA addition to the upregulation of TAM cell surface markers. TGFB1 and TGFB2 exhibited higher levels of expression in LGG tumors compared to normal brain tissue (TGFB1 mRNA expression showed a significant (*p* < 0.0001) 4.1-fold increase, and TGFB2 mRNA showed a significant (*p* < 0.0001) 2.2-fold increase in LGG tumors). TGFB ligands play a pivotal role in establishing an immunosuppressive, angiogenic, and pro-tumorigenic TME in gliomas [23,24,29,37,39,53,54], and the upregulation of these ligands in the TME provides for a compelling target for conditioning for an anti-tumor environment.

We observed increased mRNA expression of interferon-gamma receptors IFNGR1 and IFNGR2 and the downstream signaling molecules STAT1, IRF1, and IRF5, indicating an important role for IFN-γ-mediated remodeling of the TME, with implications for immunotherapies (Figure 11) [30,31,55]. The expression of IFNGR1 mRNA exhibited a significant (*p* < 0.0001) 3.1-fold increase in mRNA expression. Similarly, the mRNA expression of IFNGR2 showed a significant (*p* < 0.0001) 1.9-fold increase in mRNA expression compared to normal tissue. The expression of IRF1 mRNA revealed a significant (*p* < 0.0001) 2.11-fold increase in LGG tumors. The expression of IRF5 mRNA showed a significant (*p* < 0.0001) 3.1-fold increase in mRNA expression. STAT1 mRNA expression levels exhibited a significant (*p* < 0.0001) 2.9-fold increase comparing LGG tumors to normal brain tissue (Figure 1, Appendix A).

### 4.2. TGFB2^high^ and IFNGR2^high^ Subsets of Patients Are Over-Represented in IDHwt LGG Patients

We sought to characterize further the prognostic impact of the upregulated genes and the IDH mutational status in LGG patients, as previous studies have shown the presence of IDHwt mutational status strongly impacts OS in LGG patients, exhibiting significantly shorter progression-free survival (PFS) and overall survival (OS) compared to their IDH mutants [6,7,8,9]. In our study, the IDHwt genotype showed that the median OS of 21 months (95% CI: 17.7–25.5 months; N = 92; 50 events) was significantly shorter than that of the 421 remaining patients (median OS = 98 months (95% CI: 80–134 months; 75 events; log-rank Chi-square = 63.3, *p* < 0.0001) (Appendix A). Interestingly, there was a significant over-representation of patients with high TGFB2 and IFNGR2 mRNA expression in the IDHwt group of patients compared to the remaining patients, whereby 78 out of 92 IDH wild-type patients (85%) had high TGFB2 expression (odds ratio = 40, Fisher’s exact test, *p* < 0.0001)); and 65 out of 92 IDHwt patients (71%) had high IFNGR2 expression (odds ratio = 13.3, Fisher’s exact test, *p* < 0.0001)) (Appendix A). Application of the multivariate Cox proportional hazards model controlling for IDH mutational status and age showed increased hazard ratios for the TGFB2^high^ and IFNGR2^high^ groups of patients (HR (95% CI range) = 2.02 (1.05-3.89), *p* = 0.036 and 2.04 (1.16–3.57), *p* = 0.013 for TGFB2^high^ and IFNGR2^high^, respectively) (Appendix A). Patients with IDHwt genotype exhibited increases in HR in the models, including TGFB2 (Appendix A; HR (95% CI range) = 4.44 (1.9–10.4); *p* < 0.001) and IFNGR2 (Appendix A; HR (95% CI range) = 3.52 (1.81–6.84); *p* < 0.001). These results suggest that IDHwt designation can be used as a biomarker for LGG patients for treatment with therapies targeting the increased expression of TGFB2 and IFNGR2 mRNA. We further investigated the relationship between reduced methylation of the *TGFB2* (Figure 10A), *IFNGR2* (Figure 10B), *IRF5* (Figure 10C), *IRF1* (Figure 10D), *STAT1* (Figure 10E), and *CD276* (Figure 10F) target genes’ OS outcomes in LLG patients. All six target genes exhibited significantly worse survival outcomes at low levels of methylation (median OS times ranged from 25.5 to 48.7 months, *p* < 0.001 for all comparisons (Figure 10A–F)), suggesting that a molecular mechanism that demethylates these target genes results in worse OS outcomes in LGG patients or that mechanisms that result in hypermethylated states to improve prognosis are active in LGG tumors. Patients with low levels of *TGFB2* (92/92 = 100% versus 37/421 = 8.8%)*, IFNGR2* (64/92 = 69.6% versus 65/421 = 15.4%)*, IRF5* (52/92 = 56.5% versus 77/421 = 18.3%)*, IRF1* (68/92 = 73.9% versus 61/421 = 14.5%)*, STAT1* (55/92 = 59.8% versus 74/421 = 17.6%), and *CD276* (89/92 = 96.7% versus 40/421 = 9.5%) gene methylation were significantly over-represented in IDH-wildtype genotype patients compared to those with IDH mutations (Figure 10; *p* < 0.0001 for all comparisons).

Both IDH mutant mouse models and human glioma patients exhibit similar genetic alterations, including the hallmark IDH1 or IDH2 mutations. These mutations lead to the production of the oncometabolite 2-hydroxyglutarate (2-HG), which contributes to the inhibition of DNA demethylation enzymes and the induction of a hypermethylated phenotype known as the glioma-CpG island methylator phenotype (G-CIMP) that can impact tumor progression [60,61,62,63]. This potentially could explain the hypermethylated state of *TGFB2* and *IFGNR2* in IDHmut. Our analysis shows an over-representation of high levels of *TGFB2* and *IFNGR2/STAT1/IRF1/IRF5* gene methylations in IDHmut LGG patients. We suggest that the hypermethylated state of the *TGFB2* and *IFNGR2* pathway gene may contribute to improved prognosis among IDHmut patients.

Our analysis suggests that TGFB2’s mode of action is independent of the tumor’s IDHwt genotype, even though the IDHwt genotype is also independently negatively prognostic; it involves the co-activation of IFNGR2 pathways, eliciting downstream signaling molecules IFNGR2, IRF1, IRF5, and STAT1; CD276-positive tumor cells; and TAMs to impact OS.

### 4.3. High Levels of TGFB2, IFNGR2, and Downstream Signaling Molecules Exhibited Worse OS Outcomes than Those of the Remaining Patients

We next sought to characterize the upregulation of TGFB2 and IFNGR2 mRNA that was accompanied by highly significant correlations with worse OS in the population of LGG patients (Figure 2). Patients with high TGFB2 levels (TGFB2^high^; N = 129, 55 events) exhibited significantly shorter overall survival times (median = 27 months, 95% CI = 23.7–54.8 months) compared to those with low levels (TGFB2^low^; median = 98 months, 95% CI = 80–139 months; *p* < 0.0001), and patients with high IFNGR2 levels (IFNGR2^high^; N=129, 55 events) also had significantly shorter overall survival times (median = 34 months, 95% CI = 25.5–63.5 months) compared to those with low levels (IFNGR2^low^; median = 98 months, 95% CI = 87.5–139 months; *p* < 0.0001). Similarly, the downstream signaling molecules also exhibited significantly (*p* < 0.001 for all comparisons) shortened OS times comparing high- versus low-expression subsets of patients (Figure 3). Median OS for IRF5^high^, IRF1^high^, and STAT1^high^ groups of patients were 64 (Figure 3A), 49 (Figure 3B), and 51 months (Figure 3C), respectively, compared to the low-expression groups, which ranged from 95 to 98 months, suggesting that the activation of IFNGR2 receptor is functionally relevant and a potential therapeutic target.

### 4.4. Estimating the OS Impact of the Interaction of TGFB2 Levels and the Molecules Involved in Interferon-Gamma Receptor Activation Shows That High TGFB2 Levels Are Independent Prognostic Indicators

The levels of IFNGR2 impact the prognostic effect of TGFB2 mRNA levels in tumors of LGG patients. Patients with low levels of IFNGR2 and high levels of TGFB2 had significantly worse survival outcomes compared to those with low levels of TGFB2 (median survival times of 26 and 98 months, respectively, *p* < 0.0001). Patients with low levels of IFNGR2 and low levels of TGFB2 mRNA had better survival outcomes (median 98 months) compared to those with high levels of IFNGR2 and low levels of TGFB2 (median 64 months, *p* < 0.0001) (Figure 4). This suggests that abrogating high levels of TGFB2 is sufficient to gain survival benefits at low levels of IFNGR2. Likewise, the improved survival observed with low levels of IFNGR2 was only realized at low levels of TGFB2. Furthermore, the TGFB2 is an independent prognostic indicator for OS (HR (HR (95% CI range) = 4.07 (2.35–7.06); *p* < 0.001)), controlling for IFNGR2, age, and the interaction effect in the multivariate Cox regression model (Figure 8A). When LGG patients were partitioned according to TGFB2 and IRF5 levels, we observed that at lower expression levels of IRF5 mRNA expression, there was a significant difference between TGFB2^low^/IRF5^low^ compared to TGFB2^high^/IRF5^low^ groups of patients (105 compared to 26 months respectively; *p* < 0.0001), and this significant effect was maintained at higher levels of IRF5 expression (TGFB2^low^/IRF5^high^ versus TGFB2^high^/IRF5^high^ (93 compared to 27 months, respectively; *p* = 0.0007), suggesting the survival benefit at low levels of TGFB2 was observed regardless of the IRF5 stratification (Figure 5). The multivariate Cox model that examined the interaction effect of TGFB2 and IRF5 showed significant independent effects of the TGFB2^high^ (HR (95% CI range) = 4.48 (2.82–7.12); *p* < 0.001) and the IRF5^high^ group of patients (HR (95% CI range) = 1.94 (1.12–3.37); *p* = 0.018), suggesting that the combined inhibition of TGFB2 and IRF5 levels would result in improved survival outcomes in LGG patients (Figure 8B). Additionally, IRF1 levels do not impact the prognostic effect of TGFB2 mRNA levels, but when TGFB2 levels are low, IRF1 is a significant negative prognostic indicator for OS. Patients with low levels of IRF1 mRNA with high levels of TGFB2 had worse survival outcomes than those with low levels of TGFB2 (26 months versus 105 months, respectively; *p* < 0.0001). In addition, patients with high levels of IRF1 in combination with high levels of TGFB2 exhibited shorter OS times (median = 29 months) compared to those with high levels of IRF1 and low levels of TGFB2 mRNA (median = 64 months; *p* = 0.046). IRF1 exerted a stronger negative prognostic effect on OS than IRF5 at low levels of TGFB2 expression, whereby high levels of IRF1 exhibited significantly shorter OS times than patients expressing low levels of IRF1 (64 versus 105 months; *p* = 0.0011) (Figure 6), and the comparison of IRF5^low^/TGFB2^low^ (median = 105 months) versus IRF5^high^/TGFB2^low^ (median = 93 months) exhibited a non-significant reduced improvement of OS (Figure 5). High levels of TGFB2 demonstrated a significantly increased hazard ratio (HR (95% CI range) = 4.38 (2.67–7.17); *p* < 0.001) when applying a multivariate Cox regression model controlling for age, IRF1 levels, and an interaction effect of TGFB2 with IRF1 (Figure 8C). Given that STAT1 and IRF1 synergistically drive a large group of IFN-γ-induced genes during macrophage polarization [55], we also examined the OS of patients stratified according to TGFB2 and STAT1 levels. Patients with low levels of STAT1 mRNA had worse survival outcomes if they expressed high levels of TGFB2 compared to patients expressing low levels of STAT1 and TGFB2 (27 versus 114 months, respectively; *p* < 0.0001), and conversely, patients with low levels of TGFB2 showed that increasing the expression of STAT1 resulted in worse OS outcomes (median OS = 114 and 51 months for TGFB2^low^/STAT1^low^ versus TGFB2^low^/STAT1^high^, respectively) (Figure 7). This suggests that the levels of STAT1 impact the prognostic effect of TGFB2 mRNA levels. The multivariate Cox regression model also exhibited significantly increased hazard ratios for both TGFB2 (HR (95% CI range) = 6 (3.62–10.11); *p* < 0.001) and STAT1 (HR (95% CI range) = 2.93 (1.75–4.91); *p* < 0.001) (Figure 8D). Taken together, these results suggest that macrophage polarization in the LGG TME can be driven by interferon-gamma activation of the STAT1/IRF1/IRF5 to promote tumor progression in concert with TGFB2 levels.

### 4.5. Combinations of High Levels of TGFB2 and IFNGR2 or the Downstream IRF1/IRF5 Signaling Molecules as Targets for LGG Patients

Next, we compared the OS for patients expressing high levels of both TGFB2 and IRF5 (Figure 9A) or IRF1 (Figure 9B) to the remaining patients. Sixty-one patients expressing high levels of TGFB2 and IRF5 exhibited shorter OS times (median OS = 27 months) than the remaining patients, which was similar to the reduced OS times observed for the 74 TGFB2^high^/IRF1^high^ group of patients (median OS = 27 months). IRF5 has emerged as a promising target, and inhibitors against this transcription factor are under development. One study showed that utilizing a high-throughput screening (HTS) method of approximately 100,000 compounds identified a small-molecule compound named YE6144 that substantially inhibited the nuclear translocation of IRF5 in monocytes [64]. Also, a cell-permeable inhibitor selectively binds to IRF5 monomers (IRF5 inhibitor N5-1 (PRRVRLK)) to stabilize the inactive IRF5 monomer with sub-micromolar affinity. Preclinical treatment of NZB/W F1 mice with an inhibitor attenuated lupus pathology by reducing serum antinuclear autoantibodies, dsDNA titers, and circulating plasma cells, alleviating kidney pathology and improving survival [65].

### 4.6. CD276/B7-H3 in Combination with TGFB2 Levels as Targets for Immune Therapy in Gliomas

TGFB2 mRNA levels were positively correlated with M1 and M2 macrophage infiltration into LGG tumors (Appendix A). Examination of cell markers for TAMs expressed at high levels of TGFB2 and either CD68, MSR1, CD276, or CD86 correlation with OS outcomes in LGG patients (Figure 9C–F) showed significantly worse OS outcomes in patient groups that exhibited high levels of TGFB2 and each of the cell markers for macrophages in the TME.

In particular, there was a significant impact on OS among the groups comparing the TGFB2^high^/CD276^high^ arm of the OS curve (median OS time for 76 patients was 27 months (95% CI = 23.7–63)) with the remaining 437 patients (median OS time = 96 months; 95% CI = 78.2–120) and 89 recorded events; *p* < 0.0001) (Figure 9E). CD276/B7-H3 is a transmembrane protein that consists of four immunoglobulin domains. This protein has been identified as a tumor-associated antigen in neuroblastoma, playing a crucial role in regulating immune responses towards natural killer and T cells. B7-H3 has been shown to be expressed in several types of pediatric cancers, such as tumors of the central nervous system, sarcomas, and acute myeloid leukemia [66,67,68,69,70,71]. Focusing on Gliomas: NCT01502917, a phase 1 clinical trial tested the safety and efficacy of ^124^I-omburtamab, a monoclonal antibody labeled with ^124^I targeting CD276 delivered by convection-enhanced delivery (CED) in patients with diffuse intrinsic pontine glioma (DIPG). This trial demonstrated that CED-delivered ^124^I-omburtamab was safe and well tolerated, with no grade 4 or 5 adverse events including nausea and vomiting. Furthermore, the median OS was 15.3 months, which is longer than the typical 9–12 months for patients with diffuse intrinsic pontine glioma (DIPG) who did not receive CED. Another trial, NCT04185038, aims to test the safety and effectiveness of B7-H3-specific, chimeric antigen receptor (CAR) T cells delivered through CED into DIPG patients. The trial is accepting patients and is expected to enroll up to 25 patients with an anticipated primary completion date of May 2026. The first report of intracranial B7-H3 CAR T cells being repeatedly dosed in patients with diffuse intrinsic pontine glioma indicated preliminary findings on tolerability, the presence of CAR T cells in the CSF, and CSF cytokine elevations that suggested locoregional immune activation [72]. A phase 1 clinical trial initiated to test the safety and efficacy of SC-CAR4BRAIN (NCT05768880) targeting B7-H3, EGFR806, HER2, and IL13, a CAR T cell therapy, in children and young adults with DIPG, diffuse midline glioma (DMG), and recurrent or refractory central nervous system (CNS) tumors plans to enroll 72 patients with a primary completion date of 15 January 2028.

Our present study supports targeting the TGFB2 mRNA through inhibition or RNA interference as a potential therapeutic strategy to promote an anti-glioma immune response [49,73]. This has been demonstrated in previous studies using CED of OT101, a TGFB2-specific synthetic phosphorothioate antisense oligodeoxynucleotide (S-ODN), to treat recurrent/refractory WHO grade 3 anaplastic astrocytoma (R/R AA) patients, which resulted in a sustained partial response (PR) or complete response (CR) in 14 out of 27 patients (51.9%) [74]. The median overall survival (OS) time of 1136 days was significantly better than the temozolomide (TMZ)-treated control patient population [74]. Another study observed a significant size reduction in tumors in both GBM and AA patients after 4–11 cycles of OT101. Twenty-five percent (19/77) of the patients achieved durable objective responses, and six achieved a 100% reduction in the 3D tumor volume of the target lesions [75]. However, the multivariate modeling of mRNA expression levels impacting OS in LGG patients implicates IFN-γ-mediated remodeling of the TME via the IFNGR2/IRF1/IRF5/STAT1 axis, which also needs to be considered to improve OS outcomes. A recent study examined neuronal nitric oxide synthase (nNOS)-mediated signaling in IFN-γ-stimulated melanoma progression and the anti-melanoma effects of novel nNOS inhibitors. Co-treatment with novel nNOS small molecule inhibitors (MAC-3-190 and HH044) effectively alleviated IFN-γ-activated STAT1/3 [76]. This provides an intriguing possibility to treat LGG with a combination of OT101 and a small molecule inhibitor targeting the signaling in IFN-γ-stimulated tumor progression.

These studies’ current limitations are that they relied upon mRNA expression levels and methylation data to identify potential prognostic markers in LGG patients, which may not be correlated to protein levels or specific biological functions. Future studies looking at protein levels and signaling activities will be needed to expand on these initial findings.

## 5. Conclusions

Significant upregulation of three TAM markers, MSR1/CD204 exhibiting a 6-fold increase (*p* < 0.0001), CD86 exhibiting an 8.9-fold increase (*p* < 0.0001), and CD68 exhibiting a 15.6-fold increase in mRNA expression levels, was observed in LGG tumors. In addition, both TGFB1 and TGFB2 ligands were upregulated in these tumors compared to normal brain tissue (TGFB1 mRNA expression showed a significant (*p* < 0.0001) 4.1-fold increase, and TGFB2 mRNA showed a significant (*p* < 0.0001) 2.2-fold increase in LGG tumors), suggesting that TGFB ligands are pivotal in establishing an immunosuppressive, angiogenic, and pro-tumorigenic TME in gliomas mediated through TAMs. The complexity of the TME was further explored via the observation that increased mRNA expression of interferon-gamma receptors IFNGR1 and IFNGR2, and the downstream signaling molecules STAT1, IRF1, and IRF5, pointed to an essential role for IFN-γ mediated remodeling of the TME (Figure 11). Interestingly, a tumor-associated antigen CD276/B7-H3 mRNA expression also showed a significant (*p* < 0.0001) 4.03-fold increase in tumor tissue, providing further insights into the roles of macrophages and tumor cells in the immunosuppressive TME. Multivariate Cox proportional hazards models investigating the interaction of TGFB2 and activation of IFNGR2, STAT1, IRF1, or IRF5 showed that the prognostic impact of high mRNA levels of TGFB2 was independent of age at diagnosis and each of these signaling molecules, and furthermore, the negative prognostic impact of the signaling molecules was independent of TGFB2. Therefore, to improve OS in LGG patients, combination therapies need to be considered to target TGFB2 and IFN-γ activation. Patients with high levels of TGFB2 and IFNGR2 were over-represented by LGG patients with IDHwt mutation status, and the prognostic impact of high levels of TGFB2 or IFNGR2 observed by the increases in HR for TGFB2 and IFNGR2 were independent of IDH mutational status. Conversely, the increase in HR of IDHwt LGG patients was independent of the increases observed for high levels of TGFB2 or IFNGR2. Our results suggest that LGG patients with IDHwt and high levels of TGFB2 mRNA present a high risk for worse OS outcomes. Taken together, these results suggest that the IDHwt genotype is a negative prognostic marker, and knockdown of TGFB2 mRNA expression would improve OS outcomes independent of IDH mutational status through a mechanism that co-activates macrophages via IFNGR2 and the downstream signaling molecules IRF1, IRF5, and STAT1. Furthermore, we demonstrate that demethylation of *TGFB2, IFNGR2, IRF1, IRF5, STAT1,* and *CD276* genes results in poor prognosis and that patients with low levels of methylation of these genes are significantly over-represented in LGG IDHwt patients. The mode of action of TGFB2 is complex in the LGG TME, involving the activation of IFNGR2 pathways, tumor cells, and TAMs independent of the IDHwt genotype of tumors. TGFB2 can be directly targeted using TGFB2-specific synthetic phosphorothioate antisense oligodeoxynucleotide (OT-101) [75], which reached completion in phase 2 clinical trials in gliomas and pancreatic cancer. The data herein provide the basis for patient selection for TGFB2 therapy. This will be explored in future clinical trials with OT-101. Furthermore, IRF5 can be targeted with a small molecule compound named YE6144, an inhibitor of nuclear translocation of IRF5 in monocytes [64]. Also, a cell-permeable inhibitor that selectively binds to IRF5 monomers (IRF5 inhibitor N5-1 (PRRVRLK)) acts to stabilize IRF5 monomers with sub-micromolar affinity [65]. Combination therapies in the preclinical model and, ultimately, in clinical trials of OT-101 and IRF5 inhibitors could potentially yield improved therapeutic outcomes.

## Figures and Tables

**Figure 1 cancers-16-01202-f001:**
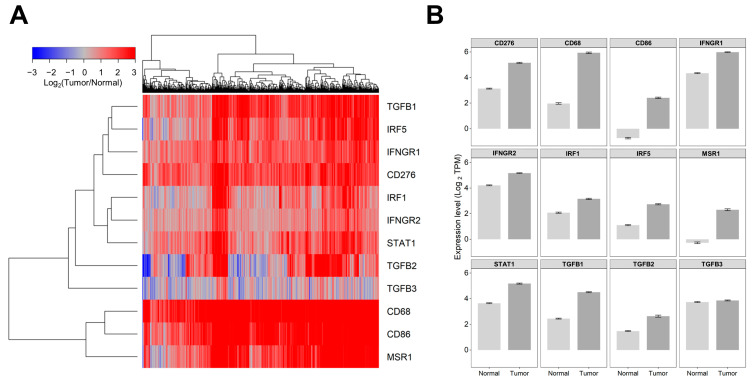
Comparing the levels of mRNA expression in normal brain tissue and tumor samples obtained from patients with LGG. We utilized log2 transformed transcripts per million (TPM) summarized RNAseq data files (https://toil-xena-hub.s3.us-east-1.amazonaws.com/download/TcgaTargetGtex_rsem_gene_tpm.gz, accessed on 25 July 2023) downloaded from the UCSC Xena web platform (https://xenabrowser.net/datapages/, accessed on 25 July 2023) to compare gene expression levels for 509 LGG patients versus 1141 brain samples from all regions of the brain. This resource reports results from the UCSC Toil RNAseq recompute compendium, which is a standardized realigned and recalculated gene and transcript expression data set for all TCGA, TARGET, and GTEx samples that enables users to contrast gene and transcript expression between TCGA “tumor” samples and corresponding GTEx “normal” samples. (**A**) Depicted is a cluster figure of the mRNA expression levels for TGFB1/2/3, interferon-gamma-activated receptors and the corresponding downstream signaling molecules (IFNGR1, IFNGR2, STAT1, IRF1, and IRF5), and markers for tumor-associated macrophages (TAMs) from LGG patients mean centered to the corresponding mRNA expression levels in normal brain tissue. The cluster figure shows the log2-transformed fold-change values (blue represents underexpression, and the red color represents overexpression in samples from LGG patients). TGFB1, IRF5, IFNGR1, and CD276 formed a co-regulated cluster, and the three TAM markers, CD68, CD86, and MSR1, formed another highly coregulated cluster. (**B**) Examination of the average expression levels of these genes (±SEM) revealed that all genes except TGFB3 (*p* = 0.1) were upregulated in tumor tissue (*p* < 0.0001).

**Figure 2 cancers-16-01202-f002:**
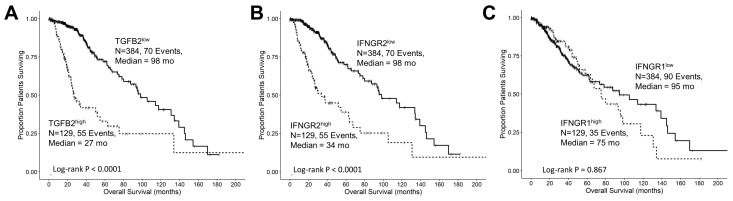
Low-grade glioma patients with high levels of TGFB2 and IFNGR2 mRNA expression exhibited significantly shorter OS times than the remaining patients. We analyzed clinical metadata and RNA sequencing data from 513 patients who were diagnosed with LGG (https://www.cbioportal.org/study/summary?id=lgg_tcga_pan_can_atlas_2018, accessed 20 April 2023). The prognostic significance was evaluated for high- and low-expressing subsets of LGG patients (25th percentile cut-off) for mRNA expression of TGFB2 (**A**), IFNGR2 (**B**), and IFNGR1 (**C**) expression in LGG patients. High levels of TGFB2 and IFNGR2 exhibited significantly shorter OS times compared to those with low levels of gene expression (*p* < 0.0001). However, the OS outcome for the IFNGR1^high^ subset of patients was not statistically different from the OS outcome in the IFNGR1^low^ subset of patients. Solid lines represent low levels of mRNA expression, and dashed lines represent high levels of mRNA expression.

**Figure 3 cancers-16-01202-f003:**
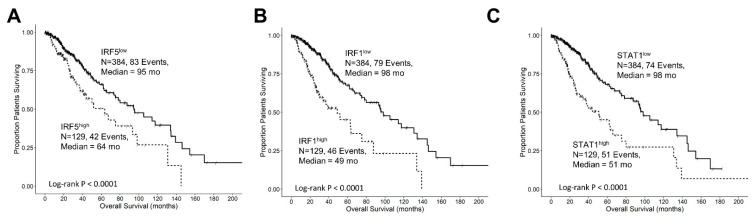
Prognostic significance of interferon-gamma receptor-activated downstream signaling molecules in LGG patients. We analyzed clinical metadata and RNA sequencing data from 513 patients who were diagnosed with LGG (https://www.cbioportal.org/study/summary?id=lgg_tcga_pan_can_atlas_2018, accessed 20 April 2023). Our goal was to determine the prognostic impact of high levels of gene expression levels on OS outcomes in these patients. High- versus low-expressing subsets of LGG patients (25th percentile cut-off) for IRF5 (**A**), IRF1 (**B**), and STAT1 (**C**) showed that high levels of mRNA expression all exhibited significantly reduced median OS times (*p* < 0.0001). Solid lines represent low levels of mRNA expression, and dashed lines represent high levels of mRNA expression.

**Figure 4 cancers-16-01202-f004:**
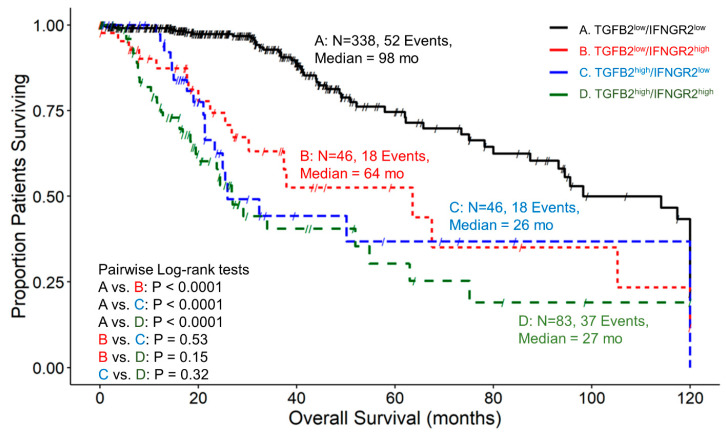
LGG patients with high levels of TGFB2 and IFNGR2 mRNA expression exhibited significantly shorter OS times than patients with low levels of TGFB2 and IFNGR2 expression. We analyzed clinical metadata and RNA sequencing data from 513 patients who were diagnosed with LGG (https://www.cbioportal.org/study/summary?id=lgg_tcga_pan_can_atlas_2018, accessed 20 April 2023). Our goal was to determine the prognostic impact of high levels of expression for both TGFB2 and IFNGR2 (upper 25th percentile for the TGFB2 and IFNGR2; TGFB2^high^/IFNGR2^high^) and OS times compared to the patients with low expression of both genes (TGFB2^low^/IFNGR2^low^; below 75th percentile for both genes) or high and low combinations of both genes (TGFB2^low^/IFNGR2^high^, TGFB2^high^/IFNGR2^low^). The curves represent TGFB2^low^/IFNGR2^low^ (A. black line), the TGFB2^low^/IFNGR2^high^ group of patients (B. redline), the TGFB2^high^/IFNGR2^low^ subset (C. blue line), and lastly, the TGFB2^high^/IFNGR2^high^ patients (D. green line).

**Figure 5 cancers-16-01202-f005:**
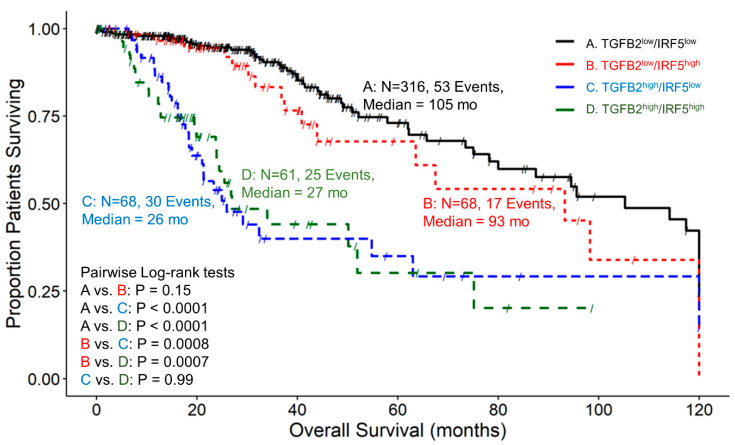
LGG patients with high levels of TGFB2 and IRF5 mRNA expression exhibited significantly shorter OS times than patients with low levels of TGFB2 and IRF5 expression. We analyzed clinical metadata and RNA sequencing data from 513 patients who were diagnosed with LGG (https://www.cbioportal.org/study/summary?id=lgg_tcga_pan_can_atlas_2018, accessed 20 April 2023). Our goal was to determine the correlation between high levels of expression for both TGFB2 and IRF5 (upper 25th percentile for the TGFB2 and IRF5; TGFB2^high^/IRF5^high^) and overall survival times compared to the patients with low expression of both genes (TGFB2^low^/IRF5^low^; below 75th percentile for both genes) or high and low combinations of both genes (TGFB2^low^/IRF5^high^, TGFB2^high^/IRF5^low^). The curves represent TGFB2^low^/IRF5^low^ (A. black line), the TGFB2^low^/IRF5^high^ group of patients (B. redline), the TGFB2^high^/IRF5^low^ subset (C. blue line), and lastly, the TGFB2^high^/IRF5^high^ patients (D. green line).

**Figure 6 cancers-16-01202-f006:**
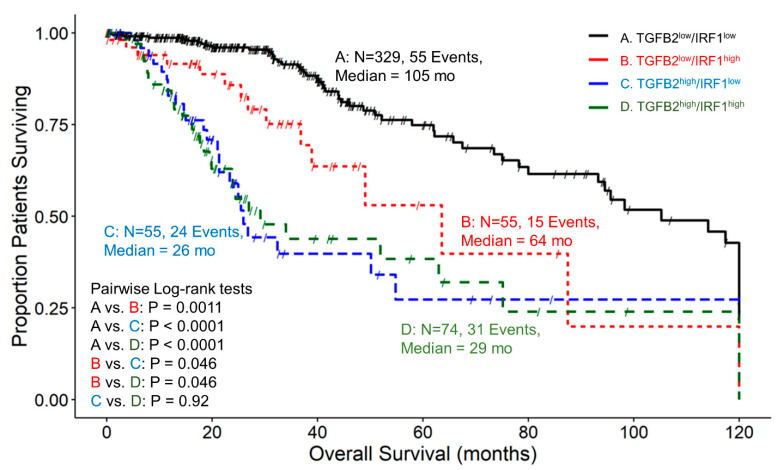
LGG patients with high levels of TGFB2 and IRF1 mRNA expression exhibited significantly shorter OS times than patients with low levels of TGFB2 and IRF1 expression. We analyzed clinical metadata and RNA sequencing data from 513 patients who were diagnosed with LGG (https://www.cbioportal.org/study/summary?id=lgg_tcga_pan_can_atlas_2018, accessed 20 April 2023). Our goal was to determine the correlation between high levels of expression for both TGFB2 and IRF1 (upper 25th percentile for the TGFB2 and IRF1; TGFB2^high^/IRF1^high^) and overall survival times compared to the patients with low expression of both genes (TGFB2^low^/IRF1^low^; below 75th percentile for both genes) or high and low combinations of both genes (TGFB2^low^/IRF1^high^, TGFB2^high^/IRF1^low^). The curves represent TGFB2^low^/IRF1^low^ (A. black line), the TGFB2^low^/IRF1^high^ group of patients (B. redline), the TGFB2^high^/IRF1^low^ subset (C. blue line), and lastly, the TGFB2^high^/IRF1^high^ patients (D. green line).

**Figure 7 cancers-16-01202-f007:**
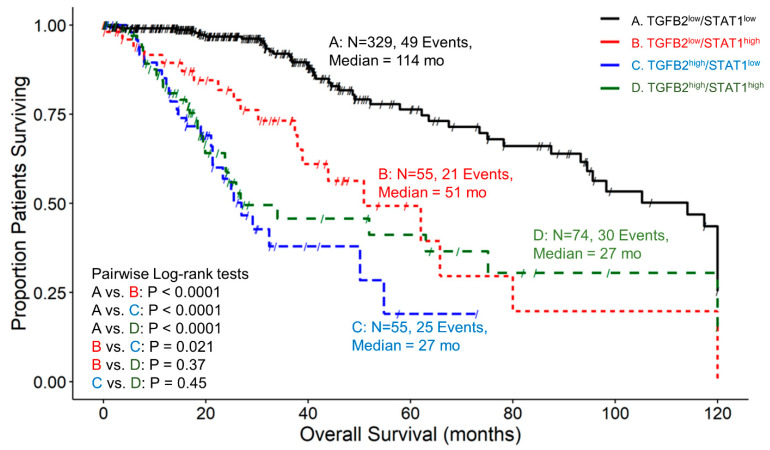
LGG patients with high levels of TGFB2 and STAT1 mRNA expression exhibited significantly shorter OS times than patients with low levels of TGFB2 and STAT1 expression. We analyzed clinical metadata and RNA sequencing data from 513 patients who were diagnosed with LGG (https://www.cbioportal.org/study/summary?id=lgg_tcga_pan_can_atlas_2018, accessed 20 April 2023). Our goal was to determine the correlation between high levels of expression for both TGFB2 and STAT1 (upper 25th percentile for the TGFB2 and STAT1; TGFB2^high^/STAT1^high^) and overall survival times compared to the patients with low expression of both genes (TGFB2^low^/STAT1^low^; below 75th percentile for both genes) or high and low combinations of both genes (TGFB2^low^/STAT1^high^, TGFB2^high^/STAT1^low^). The curves represent TGFB2^low^/STAT1^low^ (A. black line), the TGFB2^low^/STAT1^high^ group of patients (B. redline), the TGFB2^high^/STAT1^low^ subset (C. blue line), and lastly, the TGFB2^high^/STAT1^high^ patients (D. green line).

**Figure 8 cancers-16-01202-f008:**
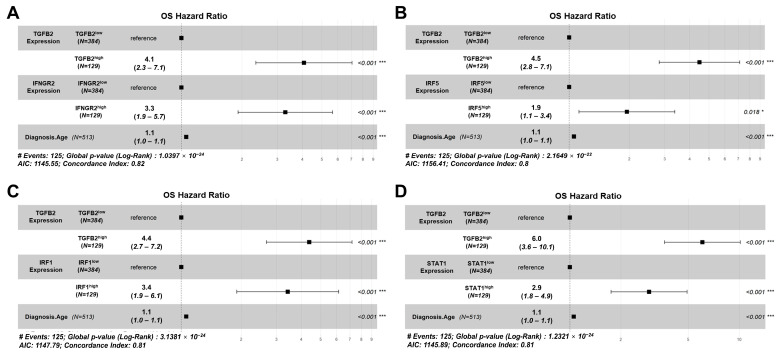
High TGFB2 mRNA expression in LGG patients was associated with higher hazard ratios when age and signaling molecules of IFN-γ receptor activation were considered in a Cox proportional hazards model that included an interaction term. Multivariate analyses were conducted to determine the impact of TGFB2 and IFN-γ receptor (IFNGR2, IRF5, IRF1, and STAT1) activation mRNA levels on OS. The Cox proportional hazards model was used to account for age at diagnosis, and an interaction term was factored into the model. (**A**) Patients in the TGFB2^high^ group and IFNGR2^high^ groups (N = 129) experienced significant increases in HR that were independent of the significant effect of the age at diagnosis (HR (95% CI range) = 1.06 (1.04–1.07); *p* < 0.001) and the interaction parameter in the model (HR (95% CI range) = 0.39 (0.18–0.85); *p* = 0.018). (**B**) According to the results of the IRF5 interaction with TGFB2 testing, the following findings were observed: Patients in the TGFB2^high^ and IRF5^high^ groups displayed a significant increase in HR that were independent of the age at diagnosis (HR (95% CI range) = 1.06 (1.04–1.07); *p* < 0.001) and the interaction term (HR (95% CI range) = 0.5 (0.23–1.08); *p* = 0.077). (**C**) LGG patients in the TGFB2^high^ expression group had a significant increase in HR, independent of the effect of IRF1^high^ expression group, age at diagnosis with an HR (95% CI range) of 1.06 (1.04–1.07) and a *p* < 0.001, and the significant effect of the interaction term with an HR (95% CI range) of 0.33 (0.15–0.73) and a *p* of 0.006. (**D**) Patients in the TGFB2^high^ expression group exhibited a significant increase in HR independent of the STAT1^high^ group of patients and independent of age at diagnosis (HR (95% CI range) = 1.05 (1.04–1.07); *p* < 0.001) and the significant effect of the interaction term (HR (95% CI range) = 0.24 (0.11–0.52); *p* < 0.001). * Denotes *p* < 0.05, and *** denotes *p* < 0.001.

**Figure 9 cancers-16-01202-f009:**
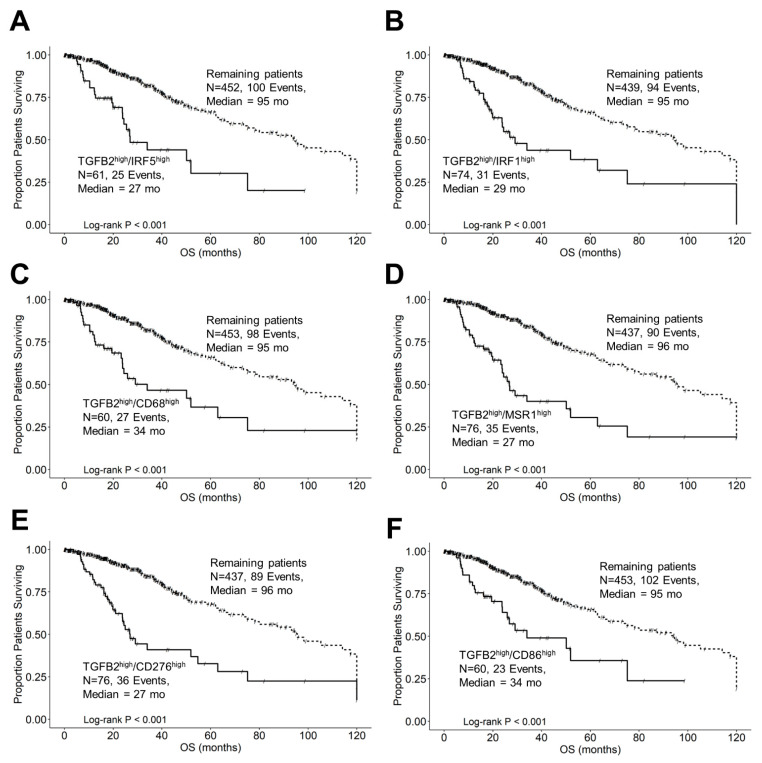
LGG patients with high levels of TGFB2 and macrophage marker mRNA expression exhibited significantly shorter OS times than the remaining patients. We analyzed clinical metadata and RNA sequencing data from 513 patients who were diagnosed with LGG (https://www.cbioportal.org/study/summary?id=lgg_tcga_pan_can_atlas_2018, accessed 20 April 2023). Our goal was to determine the correlation between high levels of expression for TGFB2, IRF1, IRF5, and macrophage markers (upper 25th percentile for the TGFB2) and overall survival times compared to the remaining patients. High levels of TGFB2 and IRF5 (**A**), IRF1 (**B**), CD68 (**C**), MSR1 (**D**), CD276 (**E**), and CD86 (**F**) mRNA expression levels all exhibited significantly reduced median OS times compared to the remaining patients (*p* < 0.0001). Solid lines represent high levels of mRNA expression for both genes, as indicated, and dashed lines represent the remaining patients.

**Figure 10 cancers-16-01202-f010:**
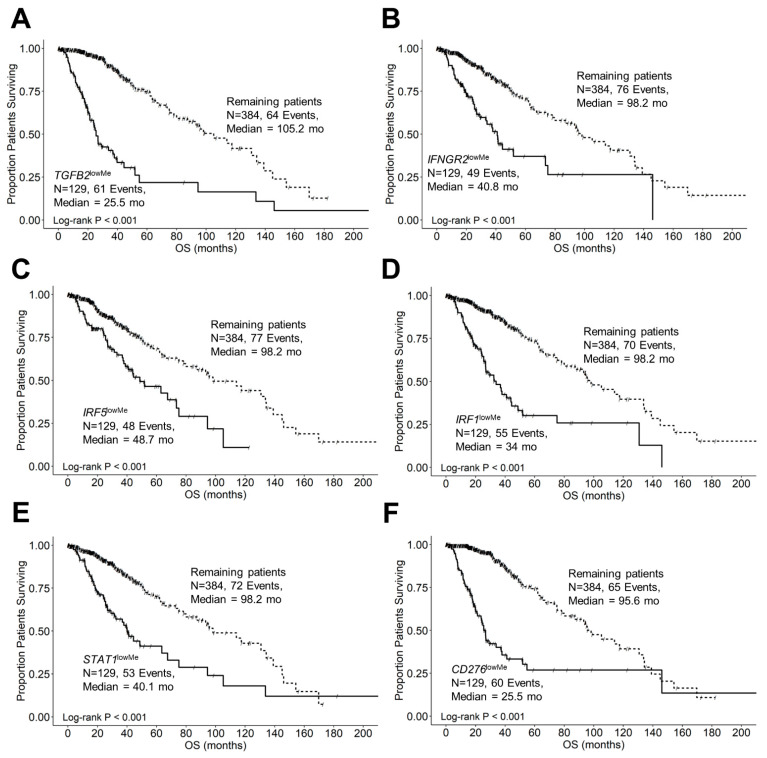
LGG patients with low levels of target gene methylation exhibited significantly shorter OS times than the remaining patients. We analyzed clinical metadata and methylation beta values obtained from HumanMethylation450 (HM450) arrays for 513 patients who were diagnosed with LGG (https://www.cbioportal.org/study/summary?id=lgg_tcga data file: “data_methylation_hm450.txt”, accessed on 13 January 2024). Our goal was to determine the correlation between reduced methylation of the target genes. Highly significant prognostic impacts were observed between reduced methylation of the target genes *TGFB2* (**A**), *IFNGR2* (**B**), *IRF5* (**C**), *IRF1* (**D**), *STAT1* (**E**), and *CD276* (**F**) (25th percentile cut-off for each gene) and worse overall survival times compared to the remaining patients. All six target genes exhibited significantly worse survival outcomes at low levels of methylation (median OS times ranged from 25.5 to 48.7 months, *p* < 0.001 for all comparisons, suggesting that low methylation levels validated the observation that increased expression of the target genes was found to be negatively prognostic in LGG patients. Patients with low levels of *TGFB2* (92/92 = 100% versus 37/421 = 8.8%)*, IFNGR2* (64/92 = 69.6% versus 65/421 = 15.4%), *IRF5* (52/92 = 56.5% versus 77/421 = 18.3%), *IRF1* (68/92 = 73.9% versus 61/421 = 14.5%), *STAT1* (55/92 = 59.8% versus 74/421 = 17.6%), and *CD276* (89/92 = 96.7% versus 40/421 = 9.5%) gene methylation were significantly over-represented in IDHwt genotype patients compared to those with IDH mutations (Fisher’s exact test, 2-tailed, *p* < 0.0001 for all comparisons). Solid lines represent low levels of gene methylation, and dashed lines represent the remaining patients.

**Figure 11 cancers-16-01202-f011:**
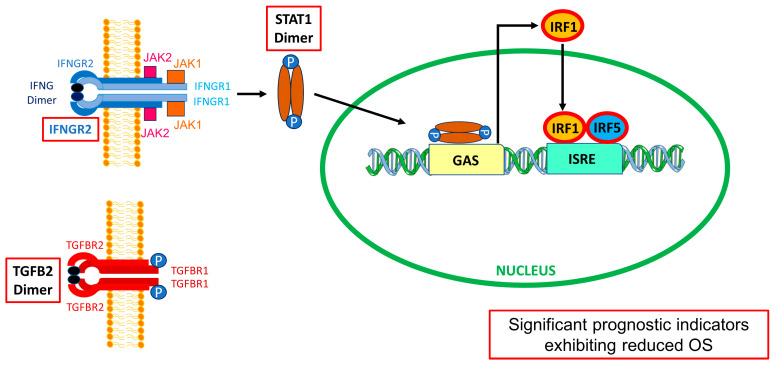
TGFB2 and IFNGR2/STAT1/IRF1/IRF5 mRNA levels prognostically impact OS outcomes in LGG patients. The schematic displays simplified models for the activation of the interferon-gamma receptor [56] and transforming growth factor beta receptor [57] on the plasma membrane by their respective ligands, IFN-γ (IFNG) and TGFB2, leading to worse OS at increased mRNA levels of IFNGR2/STAT1/IRF1/IRF5 and TGFB2 (indicated by red boxes). IFN-γ dimer signals bind to the extracellular domains of a cell surface receptor composed of an IFNGR1 and IFNGR2 tetrameric structure (blue structure). The intracellular domains of these receptors contain motifs that bind to JAK1 (orange box), associated with IFNGR1 (light blue), and JAK2 (pink box), associated with IFNGR2 (dark blue). JAK1/2 phosphorylate(s) STAT1 monomers to form STAT1 dimers (brown ellipses attached to blue circles depicting phosphorylation), which enter the nucleus to activate transcription. The STAT1 dimer binds to the gamma interferon activation site (GAS) to initiate IRF1 transcription (orange ellipse). The resulting transcription factor binds to the interferon-sensitive response element (ISRE) promoter region along with IRF5 (blue ellipse) to trigger the transcription of IFN-related genes [58]. TGFB2 dimer binds to a tetrameric receptor structure composed of TGFBR1 (2 dark red subunits) and TGFBR2 (2 red subunits). This figure was adapted from Qazi et al. (2024) [59].

## Data Availability

The authors utilized log2-transformed transcripts per million (TPM) summarized RNAseq data files (https://xenabrowser.net/datapages/ accessed on 25 July 2023) [32] to compare gene expression levels for 509 evaluable LGG patients versus 1141 brain samples from all regions of the brain. We analyzed clinical metadata and RNA sequencing-based mRNA expression data for 513 patients diagnosed with LGG ((https://www.cbioportal.org/study/summary?id=lgg_tcga_pan_can_atlas_2018; accessed 20 April 2023)). We analyzed clinical metadata and methylation beta values obtained from HumanMethylation450 (HM450) arrays for 513 patients who were diagnosed with LGG (https://www.cbioportal.org/study/summary?id=lgg_tcga data file: “data_methylation_hm450.txt”, accessed on 13 January 2024).

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
