# Peer review of "Transforming Growth Factor Beta 2 (TGFB2) mRNA Levels, in Conjunction with Interferon-Gamma Receptor Activation of Interferon Regulatory Factor 5 (IRF5) and Expression of CD276/B7-H3, Are Therapeutically Targetable Negative Prognostic Markers in Low-Grade Gliomas"

_cancers, 2024, doi:10.3390/cancers16061202_

Round 1
Reviewer 1 Report
Comments and Suggestions for Authors
The current study investigated the combined effects of TGFB2 mRNA expression and IFN-γ receptor signaling molecules (IFNGR2, IRF5, IRF1, and STAT1) on the overall survival (OS) of patients with low-grade gliomas (LGG).This study investigated the role of immune cells in low-grade gliomas (LGG) and their impact on patient survival and suggests that targeting specific molecules involved in the immunosuppressive tumor environment could be a promising strategy for improving survival in LGG patients. Overall, this article is informative, but it is very similar to the previously published study, “Transforming Growth Factor Beta 2 (TGFB2) and Interferon Gamma Receptor 2 (IFNGR2) mRNA Levels in the Brainstem Tumor Microenvironment (TME) Significantly Impact Overall Survival in Pediatric DMG Patients” by Qazi et al., Jan 2024.
1. The study employed a multivariate Cox proportional hazards model, which is a robust approach to analyzing the relationship between multiple factors and survival outcomes. The model included an interaction term to assess whether the effect of TGFB2 on survival varies depending on the levels of the signaling molecules. However, the limitation of the study is the study relies solely on mRNA expression levels as a marker of biological activity, which may not always reflect actual protein levels or function. Overall, the study provides valuable insights into the potential prognostic value of combined markers for LGG patients. However, further research is needed to validate these findings and elucidate the underlying biological mechanisms. Highlight on the limitation of the study as well.
2. I would highly recommend for the author to write the figure legends briefly and explain them in the results section.
3. If you could make a cartoon explaining the correlation of the outcomes of the hypothesis would make it easier for the readers to follow.
Author Response
Overall, this article is informative, but it is very similar to the previously published study, “Transforming Growth Factor Beta 2 (TGFB2) and Interferon Gamma Receptor 2 (IFNGR2) mRNA Levels in the Brainstem Tumor Microenvironment (TME) Significantly Impact Overall Survival in Pediatric DMG Patients” by Qazi et al., Jan 2024.
A previous study on pediatric DMG patients was confirmed in adult low-grade gliomas in this study. Both studies found similar and opposite prognosis impacts of IFNGR2 and TGFB2, suggesting a critical role of the TME and associated immune cells and molecules in mediating the prognostic outcomes.
- The study employed a multivariate Cox proportional hazards model, which is a robust approach to analyzing the relationship between multiple factors and survival outcomes. The model included an interaction term to assess whether the effect of TGFB2 on survival varies depending on the levels of the signaling molecules. However, the limitation of the study is the study relies solely on mRNA expression levels as a marker of biological activity, which may not always reflect actual protein levels or function. Overall, the study provides valuable insights into the potential prognostic value of combined markers for LGG patients. However, further research is needed to validate these findings and elucidate the underlying biological mechanisms. Highlight on the limitation of the study as well.
We appreciate the reviewer's concern that the study relies on mRNA expression levels and may not relate to protein levels or function. We have added the following section to the end of the discussion:
“These studies' current limitations are that they relied upon mRNA expression levels and methylation data to identify potential prognostic markers in LGG patients, which may not be correlated to protein levels or specific biological functions. Future studies looking at protein levels and signaling activities will be needed to expand on these initial findings.”
- I would highly recommend for the author to write the figure legends briefly and explain them in the results section.
We thank the reviewer for this suggestion and have revised the figure legends accordingly and provided more details in the results section.
- If you could make a cartoon explaining the correlation of the outcomes of the hypothesis would make it easier for the readers to follow.
We thank the reviewer for this suggestion, which will indeed make the manuscript more accessible for the reader to follow. We have now included Figure 11 in the revised manuscript.
Reviewer 2 Report
Comments and Suggestions for Authors
Low-grade gliomas (LGG) are characterized by a reduced number of immune cells infiltrating the tumor, requiring immune-boosting therapies to release the immunosuppressive environment of the tumor. A bioinformatics-led study was organized to evaluate the prognostic impact of cell surface receptor markers for tumor-associated macrophages (TAM) that produce Transforming Growth (TGFB) ligands and Interferon-gamma receptor activation to maintain the tumor micro- environment in an immunosuppressed state. Detailed investigation of messenger RNA levels and their impact on survival in LGG patients demonstrated that low mRNA levels of a specific TGFB ligand, TGFB2, and the receptor along with the signaling molecules from interferon-gamma receptor activation (IFNGR2, IRF1, IRF5, and STAT1) result in improved survival of LGG patients. LGG patients expressing high levels of TGFB2 and IFNGR2 are over-represented in IDH (isocitrate dehydrogenase) wild-type tumor samples, suggesting that TGFB2 and IFNGR2 mRNA can be therapeutically targeted in these high-risk patients. Significant up- regulation of mRNA expression levels for three TAM markers, MSR1/CD204, CD86, and CD68, suggested that TGFB2 is pivotal in establishing a pro-tumorigenic tumor microenvironment in gliomas mediated through TAMs. A tumor-associated antigen CD276/B7-H3 mRNA expression is increased in tumor tissue, giving further insights into the roles of macrophages and tumor cells in the immunosuppressive TME. Therefore, to improve OS in LGG patients, combination therapies must target TGFB2 and IFN-γ activation via immune therapies against CD276/B7-H3.
- This paper looks at a possible explanation for the lack of an immune response against low grade gliomas.
- A new finding from this study is that upon investigation of messenger RNA levels and their impact on survival in LGG patients it was demonstrated that low mRNA levels of a specific TGFB ligand, TGFB2, and the receptor along with the signaling molecules from interferon-gamma receptor activation (IFNGR2, IRF1, IRF5, and STAT1) result in improved survival of LGG patients.
- The methodology is good.
-
The conclusions are somewhat speculative. In particular it is unclear how TGFB2 and IFNGR2 mRNA therapeutic targeting in these high-risk patients can be achieved and result in an improvement in outcome of these patients.
In summary this is a lot of interesting data, but the clinical applications are speculative. There are mouse models for low grade gliomas and perhaps they can be used to test the theories developed in this paper.
This is a somewhat complex paper with potential clinical applications.
Comments on the Quality of English LanguageSome minor changes could be made.
Author Response
This paper looks at a possible explanation for the lack of an immune response against low grade gliomas.
A new finding from this study is that upon investigation of messenger RNA levels and their impact on survival in LGG patients it was demonstrated that low mRNA levels of a specific TGFB ligand, TGFB2, and the receptor along with the signaling molecules from interferon-gamma receptor activation (IFNGR2, IRF1, IRF5, and STAT1) result in improved survival of LGG patients.
The methodology is good.
The conclusions are somewhat speculative. In particular it is unclear how TGFB2 and IFNGR2 mRNA therapeutic targeting in these high-risk patients can be achieved and result in an improvement in outcome of these patients.
We have taken the reviewer’s assertion into account and modified the conclusions to highlight that, at present, we can target TGFB2 mRNA expression but will require combinations with other treatment modalities to be applied to LGG tumors.
We have now included this statement at the end of the conclusions.
“TGFB2 can be directly targeted using TGFB2-specific synthetic phosphorothioate anti-sense oligodeoxynucleotide (OT-101) [73], which completed phase 2 clinical trials in gliomas and pancreatic cancer. The data from here provide the basis for patient selection for TGFB2 therapy. This will be explored in future clinical trials with OT-101. Furthermore, IRF5 can be targeted with a small-molecule compound named YE6144, an inhibitor of nuclear translocation of IRF5 in monocytes [60]. Also, a cell-permeable inhibitor selectively binds to IRF5 monomers (IRF5 inhibitor N5-1 (PRRVRLK) acts to stabilize IRF5 monomers with sub-micromolar affinity [61]. Combination therapies in the preclinical model and, ultimately, in clinical trials of OT-101 and IRF5 inhibitors could potentially yield improved therapeutic outcomes.”
In summary this is a lot of interesting data, but the clinical applications are speculative. There are mouse models for low grade gliomas and perhaps they can be used to test the theories developed in this paper.
We agree with the reviewer and are planning to perform mouse studies to expand on these results.
This is a somewhat complex paper with potential clinical applications.
Reviewer 3 Report
Comments and Suggestions for Authors
Trieu et al. investigate the role of transforming growth factor beta-2 (TGFB2) and the presence of active signaling molecules (IFNGR2, IRF1, IRF5 and STAT1) in modulating the immune microenvironment in the setting of low-grade gliomas, finding that the presence of their unmethylated genes allows for their expression and reducing immune targeting of low-grade tumors, noting that this is present in greater amounts in lower prognosis (ie IDH wild type) low-grade gliomas. The supposition is that targeting TGFB2 and associated signaling molecules would allow for improved tumor control. Findings were that low levels of gene methylation in a variety of genes, including TGFB2, IFNGR2, IRF1, IRF5 and STAT1 among others were associated with worsening outcome, supporting these as targets for tumor therapy.
ABSTRACT: The abstract adequately summarized the contents of the submission.
INTRODUCTION: the introduction provided excellent background material to the genetic mutations potentially present in low-grade gliomas and their targeting, and provides a hypothesis for the role of tumor associated macrophages and the subtypes that suppress the immune environment in low-grade gliomas.
MATERIALS AND METHODS: No issues in experimental design
RESULTS: Results were relatively easy to understand and statistics were appropriately applied.
DISCUSSION: The discussion explained the results in an understandable manner and explain the conclusions logically.
CONCLUSIONS: The conclusion section adequately summarizes the findings. The authors could explain how this pathway would be targeted. The authors should indicate if they are aware of any substances that may be used to target this pathway. Additionally, the authors emphasized that IDH wild type is a negative prognostic marker in low-grade glioma, something that is known from other data. The authors should address if there is an actual relationship, causal or otherwise, between the IDH mutation and the methylations seen in IDH wild type tumors?
REFERENCES: All references are appropriate and up-to-date.
FIGURES AND TABLES: All complement/supplement the data provided and do not duplicate information provided in the narrative section of the report.
Author Response
Comments and Suggestions for Authors
Findings were that low levels of gene methylation in a variety of genes, including TGFB2, IFNGR2, IRF1, IRF5 and STAT1 among others were associated with worsening outcome, supporting these as targets for tumor therapy.
We appreciate the reviewer pointing out that the methylation data supported the targets described by the mRNA expression correlation with OS outcomes.
ABSTRACT: The abstract adequately summarized the contents of the submission.
We appreciate the abstract satisfied the reviewer.
INTRODUCTION: the introduction provided excellent background material to the genetic mutations potentially present in low-grade gliomas and their targeting, and provides a hypothesis for the role of tumor associated macrophages and the subtypes that suppress the immune environment in low-grade gliomas.
We appreciate the reviewer’s comment on the introduction.
MATERIALS AND METHODS: No issues in experimental design
We appreciate the methods section was to the satisfaction of the reviewer.
RESULTS: Results were relatively easy to understand and statistics were appropriately applied.
We thank the reviewer for this comment.
DISCUSSION: The discussion explained the results in an understandable manner and explain the conclusions logically.
We thank the reviewer for this comment.
CONCLUSIONS: The conclusion section adequately summarizes the findings. The authors could explain how this pathway would be targeted. The authors should indicate if they are aware of any substances that may be used to target this pathway.
We have now included the following summary in the conclusions section to address how the pathway could be targeted.
“TGFB2 can be directly targeted using TGFB2-specific synthetic phosphorothioate anti-sense oligodeoxynucleotide (OT-101) [73], which completed phase 2 clinical trials in gliomas and pancreatic cancer. The data from here provide the basis for patient selection for TGFB2 therapy. This will be explored in future clinical trials with OT-101. Furthermore, IRF5 can be targeted with a small-molecule compound named YE6144, an inhibitor of nuclear translocation of IRF5 in monocytes [60]. Also, a cell-permeable inhibitor selectively binds to IRF5 monomers (IRF5 inhibitor N5-1 (PRRVRLK) acts to stabilize IRF5 monomers with sub-micromolar affinity [61]. Combination therapies in the preclinical model and, ultimately, in clinical trials of OT-101 and IRF5 inhibitors could potentially yield improved therapeutic outcomes.”
Additionally, the authors emphasized that IDH wild type is a negative prognostic marker in low-grade glioma, something that is known from other data.
We concur with the reviewer that it is well-known that IDHwt is a negative prognostic marker. We extended this observation using the multivariate Cox proportional hazards model that included TGFB2 mRNA expression levels and IDH mutational status as co-variates (Figure S2). This model exhibited independent prognostic impact for TGFB2 and IDH mutational status (HR = 2.02 for high TGFB2, and HR = 4.44 for IDHwt). This independent effect of TGFB2 mRNA may be driven by the demethylated state of the TGFB2 gene as observed in Figure 10A that showed low levels of TGFB2 methylation leads to worse prognosis and over-representation of the IDHwt genotype in the subset of these patients with low TGFB2 gene methylation (Patients with low levels of TGFB2 methylation were significantly over-represented in the patients with IDHwt genotype (92/92 = 100%) compared to 37 TGFB2lowMe patients with IDH mutations (37/421 = 8.8%; Fishers Exact test, 2-tailed, P<0.0001)). We suggest that patients with IDH mutation maintain the hypermethylated state of the TGFB2 gene contributing to improved OS.
The authors should address if there is an actual relationship, causal or otherwise, between the IDH mutation and the methylations seen in IDH wild type tumors?
LGG tumors are classified either as harboring IDH mutations or IDHwt. Intriguingly, these IDH mutations lead to the production of the oncometabolite 2-hydroxyglutarate (2-HG) that inhibits DNA demethylation and induces a hypermethylated phenotype known as the glioma-CpG island methylator phenotype (G-CIMP). This potentially could explain the hypermethylated state of TGFB2 and IFGNR2 in IDHmut. Our analysis shows an over-representation of high levels of TGFB2 and IFNGR2/STAT1/IRF1/IRF5 gene methylations in IDHmut LGG patients. We suggest that the hypermethylated state of the TGFB2 and IFNGR2 pathway gene may contribute to improved prognosis among IDHmut patients.
We have now modified the discussion in Section 4.2 to highlight this observation.
“Both IDH mutant mouse models and human glioma patients exhibit similar genetic alterations, including the hallmark IDH1 or IDH2 mutations. These mutations lead to the production of the oncometabolite 2-hydroxyglutarate (2-HG), which contributes to the inhibition of DNA demethylation enzymes and the induction of a hypermethylated phenotype known as the glioma-CpG island methylator phenotype (G-CIMP) that can impact tumor progression [56–59]. This potentially could explain the hypermethylated state of TGFB2 and IFGNR2 in IDHmut. Our analysis shows an over-representation of high levels of TGFB2 and IFNGR2/STAT1/IRF1/IRF5 gene methylations in IDHmut LGG patients. We suggest that the hypermethylated state of the TGFB2 and IFNGR2 pathway gene may contribute to improved prognosis among IDHmut patients.”
REFERENCES: All references are appropriate and up-to-date.
We appreciate the reviewer's attention to this section.
FIGURES AND TABLES: All complement/supplement the data provided and do not duplicate information provided in the narrative section of the report.
We appreciate the reviewer's attention to this section.